# Correlation detection as a stimulus computable account for audiovisual perception, causal inference, and saliency maps in mammals

Cesare V Parise*

Department of Psychology, Institute of Population Health, Faculty of Health and Life Sciences, University of Liverpool, Liverpool, United Kingdom

## eLife Assessment

This **important** study evaluates a model for multisensory correlation detection, focusing on the detection of correlated transients in visual and auditory stimuli. Overall, the experimental design is sound and the evidence is **compelling**. The synergy between the experimental and theoretical aspects of the article is strong, and the work will be of interest to both neuroscientists and psychologists working in the domain of sensory processing and perception

**\*For correspondence:**
cesare.parise@liverpool.ac.uk

**Competing interest:** The author declares that no competing interests exist.

**Abstract** Animals excel at seamlessly integrating information from different senses, a capability critical for navigating complex environments. Despite recent progress in multisensory research, the absence of stimulus-computable perceptual models fundamentally limits our understanding of how the brain extracts and combines task-relevant cues from the continuous flow of natural multisensory stimuli. Here, we introduce an image- and sound-computable population model for audiovisual perception, based on biologically plausible units that detect spatiotemporal correlations across auditory and visual streams. In a large-scale simulation spanning 69 psychophysical, eye-tracking, and pharmacological experiments, our model replicates human, monkey, and rat behaviour in response to diverse audiovisual stimuli with an average correlation exceeding 0.97. Despite relying on as few as 0–4 free parameters, our model provides an end-to-end account of audiovisual integration in mammals—from individual pixels and audio samples to behavioural responses. Remarkably, the population response to natural audiovisual scenes generates saliency maps that predict spontaneous gaze direction, Bayesian causal inference, and a variety of previously reported multisensory illusions. This study demonstrates that the integration of audiovisual stimuli, regardless of their spatiotemporal complexity, can be accounted for in terms of elementary joint analyses of luminance and sound level. Beyond advancing our understanding of the computational principles underlying multisensory integration in mammals, this model provides a bio-inspired, general-purpose solution for multimodal machine perception.

## Introduction

Perception in natural environments is inherently multisensory. For example, during speech perception, the human brain integrates audiovisual information to enhance speech intelligibility, often beyond awareness. A compelling demonstration of this is the McGurk illusion (*McGurk and MacDonald, 1976*), where the auditory perception of a syllable is altered by mismatched lip movements. Likewise,

audiovisual integration plays a critical role in spatial localization, as illustrated by the ventriloquist illusion (*Stratton, 1897*), where perceived sound location shifts toward a synchronous visual stimulus.

Extensive behavioural and neurophysiological findings demonstrate that audiovisual integration occurs when visual and auditory stimuli are presented in close spatiotemporal proximity (i.e. the spatial and temporal determinants of multisensory integration; *Stein, 2012*, *Stein and Stanford, 2008*). When redundant multisensory information is integrated, the resulting percept is more reliable (*Ernst and Banks, 2002*) and salient (*Talsma et al., 2010*). Various models have successfully described how audiovisual integration unfolds across time and space (*Alais and Burr, 2004*, *Körding et al., 2007*, *Magnotti et al., 2013*, *Yarrow et al., 2023*)–often within a Bayesian Causal Inference framework, where the system determines the probability that visual and auditory stimuli have a common cause and weighs the senses accordingly. This is the case for the detection of spatiotemporal discrepancies across the senses, or susceptibility to phenomena such as the McGurk or Ventriloquist illusions (*Körding et al., 2007*, *Magnotti et al., 2013*, *Magnotti and Beauchamp, 2017*).

Prevailing theoretical models of multisensory integration typically operate at what *Marr, 1982* termed the computational level: they describe what the system is trying to achieve (e.g. obtain precise sensory estimates). However, these models are not stimulus-computable. That is, rather than analysing raw auditory and visual input directly, they rely on experimenter-defined, low-dimensional abstractions of the stimuli (*Alais and Burr, 2004*, *Körding et al., 2007*, *Magnotti et al., 2013*, *Yarrow et al., 2023*, *Magnotti and Beauchamp, 2017*)—such as the asynchrony between sound and image, expressed in seconds (*Magnotti et al., 2013*, *Yarrow et al., 2023*), or spatial location (*Alais and Burr, 2004*, *Körding et al., 2007*). As a result, they solve a fundamentally different task than real perceptual systems, which must infer such properties from the stimuli themselves—from dynamic patterns of pixels and audio samples—without access to ground-truth parameters.

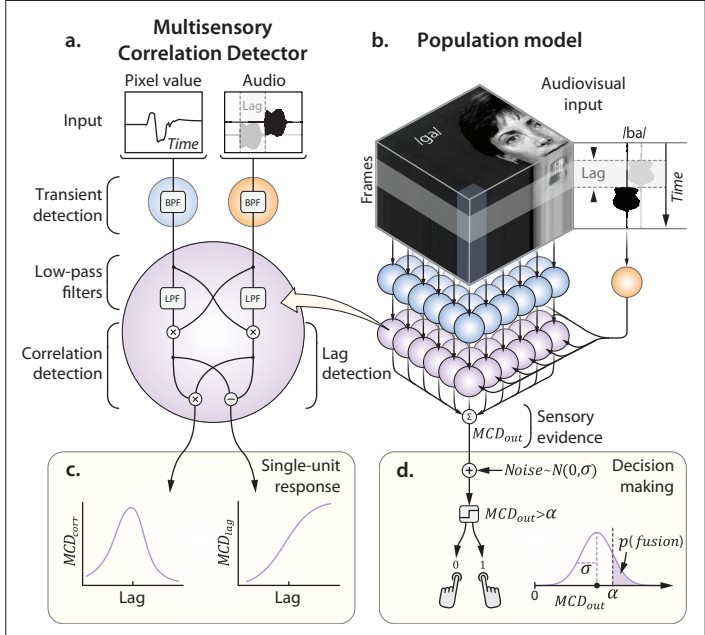

**Figure 1.** The Multisensory Correlation Detector (MCD) population model. (**A**) Schematic representation of a single MCD unit. The input visual signal represents the intensity of a pixel (mouth area) over time, while the audio input is the soundtrack (the syllable /ba/). The gray soundtrack represents the experimental manipulation of AV lag, obtained by delaying one sense with respect to the other. BPF and LPF indicate band-pass and low-pass temporal filters, respectively. (**C**) shows how single-unit responses vary as a function of cross-modal lag. (**B**) represents the architecture of the MCD population model. Each visual unit (in blue) receives input from a single pixel, while the auditory unit receives as input the intensity envelope of the soundtrack (mono audio; see *Figure 4A* for a version of the model capable of receiving spatialized auditory input). Sensory evidence is then integrated over time and space for perceptual decision-making, a process in which the two model responses are weighted, summed, corrupted with additive Gaussian noise, and compared to a criterion to generate a forced-choice response (**D**).

From Marr's perspective, what is missing is an account at the algorithmic level: a concrete description of the stimulus-driven representations and operations that could give rise to the observed computations.

Despite their clear success in accounting for behaviour in simple, controlled conditions, current models remain silent on how perceptual systems extract, process, and combine task-relevant information from the continuous and structured stream of audiovisual signals that real-world perception entails. This omission is critical: audiovisual perception involves the continuous analysis of images and sounds; hence, models that do not operate on the stimuli cannot provide a complete account of perception. Only a few models can process elementary audiovisual stimuli (*Parise and Ernst, 2016*, *Cuppini et al., 2017*), none can tackle the complexity of natural audiovisual input. Currently, there are no stimulus-computable models (*Burge, 2020*) for multisensory perception that can take as input natural audiovisual data, like movies. This study explores how behaviour consistent with mammalian multisensory perception emerges from low-level analyses of natural auditory and visual signals.

In an image- and sound-computable model, visual and auditory stimuli can be represented as patterns in a three-dimensional space, where *x* and *y* are the two spatial dimensions, and *t is* the temporal dimension. An instance of such a three-dimensional diagram for the case of audiovisual speech is shown in *Figure 1B* (top): moving lips generate patterns of light that vary in sync with the sound. In such a representation, audiovisual correspondence can be detected by a local correlator (i.e. multiplier), that operates across space, time, and the senses (*Parise et al., 2012*). In previous studies, we proposed a biologically plausible solution to detect temporal correlation across the senses (*Figure 1A*; *Parise and Ernst, 2016*; *Pesnot Lerousseau et al., 2022*; *Parise and Ernst, 2025*; *Horsfall et al., 2021*). Here, we will illustrate how a population of multisensory correlation detectors can take real-life footage as input and provide a comprehensive bottom-up account for multisensory integration in mammals, encompassing its temporal, spatial, and attentional aspects.

The present approach posits the existence of elementary processing units, the Multisensory Correlation Detectors (MCD; *Parise and Ernst, 2025*), each integrating time-varying input from unimodal transient channels through a set of temporal filters and elementary operations (*Figure 1A*, see Methods). Each unit returns two outputs, representing the temporal correlation and order of incoming visual and auditory signals (*Figure 1C*). When arranged in a two-dimensional lattice (*Figure 1B*), a population of MCD units is naturally suited to take movies (e.g. dynamic images and sounds) as input, hence capable to process any stimuli used in previous studies in audiovisual integration. Given that the aim of this study is to provide an account for multisensory integration in biological system, the benchmark of our model is to reproduce observers' behaviour in carefully controlled psychophysical and eye-tracking experiments. Emphasis will be given to studies using natural stimuli, which despite their manifest ecological value, simply cannot be handled by alternative models. Among them, particular attention will be dedicated to experiments involving speech, perhaps the most representative instance of audiovisual perception, and sometimes claimed to be processed via dedicated mechanisms in the human brain.

## Results

We tested the performance of our population model on three main aspects of audiovisual perception. The first concerns the temporal determinants of multisensory integration, primarily investigating how subjective audiovisual synchrony and integration depend on the physical lag across the senses. The second addresses the spatial determinants of audiovisual integration, focusing on the combination of visual and acoustic cues for spatial localization. The third one involves audiovisual attention and examines how gaze behaviour is spontaneously attracted to audiovisual stimuli even in the absence of explicit behavioural tasks. While most of the literature on audiovisual psychophysics involves human participants, in recent years monkeys and rats have also been trained to perform the same behavioural tasks. Therefore, to generalize our approach, whenever possible, we simulated experiments involving all available animal models.

### Temporal determinants of audiovisual integration in humans and rats

Classic experiments on the temporal determinants of audiovisual integration usually manipulate the lag between the senses and assess the perception of synchrony, temporal order, and audiovisual

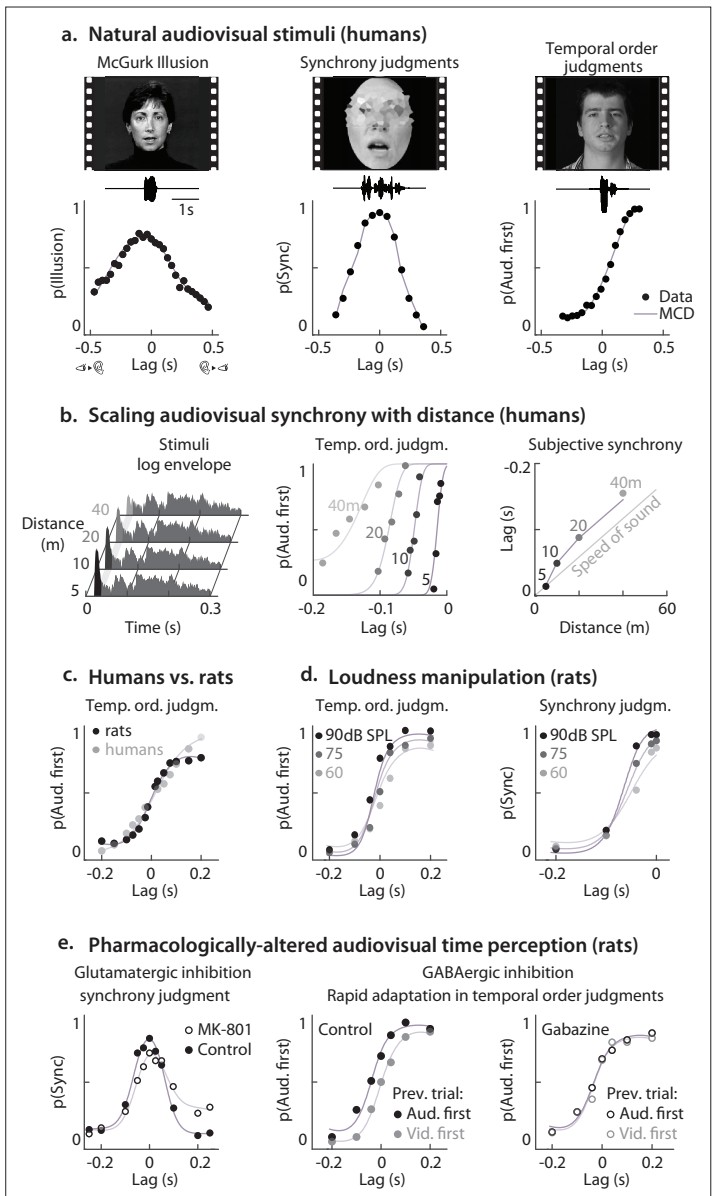

**Figure 2.** Natural audiovisual stimuli and psychophysical responses. (**A**) Stimuli (still frame and soundtrack) and psychometric functions for McGurk illusion (***van Wassenhove et al., 2007***), synchrony judgments (***Lee and Noppeney, 2011***), and temporal order judgments (***Vroomen and Stekelenburg, 2011***). In all panels, dots correspond to empirical data, lines to Multisensory Correlation Detectors (MCD) responses; negative lags represent vision first. (**B**) Stimuli and results of ***Alais and Carlile, 2005***. The left panel displays the envelopes of auditory stimuli (clicks) recorded at different distances in a reverberant environment (the Sydney Opera House). While the reverberant portion of the sound is identical across distances, the intensity of the direct sound (the onset) decreases with depth. As a result, the centre of mass of the envelopes shifts rightward with increasing distance. The central panel shows empirical and predicted psychometric functions for the various distances. The four curves were fitted using the same decision-making parameters, so that the separation between the curves results purely from the operation of the MCD. The lag at which sound and light appear synchronous (point of subjective synchrony) scales with distance at a rate approximately matching the speed of sound (right panel). The dots in the right panel display the point of subjective synchrony (estimated separately for each curve), while the jagged line is the model prediction. (**C**) shows temporal order judgments for clicks and flashes from both rats and human observers (***Mafi et al., 2022***). Rats outperform humans at short lag, and vice-versa. (**D**) Rats' temporal order and synchrony judgments for flashes and clicks of varying intensity (***Schormans and Allman, 2018***). Note that in the synchrony judgment task only the left flank of the psychometric curve (video-lead lags) was sampled. Importantly the tree curves in each task were fitted using the same decision-making parameters, so that the

*Figure 2 continued on next page*

*Figure 2 continued*

MCD alone accounts for the separation between the curves. (**E**) Pharmacologically-induced changes in rats' audiovisual time perception. Left: Glutamatergic inhibition (MK-801 injection) leads to asymmetric broadening of the psychometric functions for simultaneity judgments. Right: GABA inhibition (Gabazine injection) abolishes rapid temporal adaptation, so that psychometric curves do not change based on the lag of the previous trials (as they do in controls) (*Schormans and Allman, 2023*). All pharmacologically-induced changes in audiovisual time perception can be accounted for by changes in the decision-making process, with no need to postulate changes in low-level temporal processing.

The online version of this article includes the following figure supplement(s) for figure 2:

**Figure supplement 1.** Temporal determinants of multisensory integration in humans.

**Figure supplement 2.** Temporal determinants of multisensory integration in rats.

**Figure supplement 3.** Simultaneity judgment for impulse stimuli in humans, individual observer data (*Yarrow et al., 2023*).

**Figure supplement 4.** Simultaneity and temporal order judgments for step stimuli (*Parise and Ernst, 2025*).

**Figure supplement 5.** Simultaneity judgment for periodic stimuli (*Parise and Ernst, 2025*).

speech integration (as measured in humans with the McGurk illusion, see Video 1) through psycho-physical forced-choice tasks (*Venezia et al., 2016*, *Vroomen and Keetels, 2010*; *Parise et al., 2025*). Among them, we obtained both the audiovisual footage and the psychophysical data from 43 experiments in humans that used ecological audiovisual stimuli (real-life recordings of, e.g. speech and performing musicians, *Figure 2A*, *Figure 2—figure supplement 1* and *Supplementary file 1*, for the inclusion criteria, see Methods): 27 experiments were simultaneity judgments (*van Wassenhove et al., 2007*; *Lee and Noppeney, 2011*; *Vroomen and Stekelenburg, 2011*; *Magnotti et al., 2013*; *Roseboom and Arnold, 2011*; *Yuan et al., 2014*; *Ikeda and Morishita, 2020*; *van Laarhoven et al., 2019*; *Lee and Noppeney, 2014*) , 10 temporal order judgments (*Vroomen and Stekelenburg, 2011*, *Freeman et al., 2013*), six others assessed the McGurk effect (*van Wassenhove et al., 2007*, *Yuan et al., 2014*, *Freeman et al., 2013*).

For each of the experiments, we can feed the stimuli to the model (*Figure 1B and D*), and compare the output to the empirical psychometric functions (*Equation 10*, for details see Methods) (*Parise and Ernst, 2016*, *Pesnot Lerousseau et al., 2022*; *Parise and Ernst, 2025*; *Horsfall et al., 2021*). Results demonstrate that a population of MCDs can broadly account for audiovisual temporal perception of ecological stimuli, and near-perfectly (rho = 0.97) reproduces the empirical psychometric functions for simultaneity judgments, temporal order judgments, and the McGurk effect (*Figure 2A*, *Figure 2—figure supplement 1*). To quantify the impact of the low-level properties of the stimuli on the performance of the model, we ran a permutation test, where psychometric functions were predicted from mismatching stimuli (see Methods). The psychometric curves predicted from the matching stimuli provided a significantly better fit than mismatching stimuli (p<0.001, see *Figure 2—figure supplement 1K*). This demonstrates that our model captures the subtle effects of how individual features affect observed responses, and it highlights the role of low-level stimulus properties on multisensory perception. All analyses performed so far relied on psychometric functions averaged across observers; individual observer analyses are included in the *Figure 2—figure supplements 3–5*.

When estimating the perceived timing of audiovisual events, it is important to consider the different propagation speeds of light and sound, which introduce audio lags that are proportional to the observer's distance from the source (*Figure 2B*, right). Psychophysical temporal order judgments demonstrate that, to compensate for these lags, humans scale subjective audiovisual synchrony with distance (*Figure 2B*; *Alais and Carlile, 2005*). This result has been interpreted as evidence that humans exploit auditory spatial cues, such as the direct-to-reverberant energy ratio (*Figure 2B*, left), to estimate the distance of the sound source and adjust subjective synchrony by scaling distance estimates by the speed of sound (*Alais and Carlile, 2005*). When presented with the same stimuli, our model also predicts the observed shifts in subjective simultaneity (*Figure 2B*, centre). However, rather than relying on explicit spatial representations and physics simulations, these shifts emerge from elementary analyses of natural audiovisual signals. Specifically, in reverberant environments, the intensity of the direct portion of a sound increases with source proximity, while the reverberant

component remains constant. As a result, the envelopes of sounds originating close to the observers are more front-heavy than distant sounds (*Figure 2B*, left). These are low-level acoustic features that the lag detector of the MCD is especially sensitive to, thereby providing a computational shortcut to explicit physics simulations. A Matlab implementation of this simulation is included in *Source code 1*.

In recent years, audiovisual timing has been systematically studied also in rats (*Mafi et al., 2022*, *Schormans and Allman, 2018*, *Al Youzbaki et al., 2023*, *Schormans and Allman, 2023*, *Schormans et al., 2016*, *Paulcan et al., 2023*), generally using minimalistic stimuli (such as clicks and flashes), and under a variety of manipulations of the stimuli (e.g. loudness) and pharmacological interventions (e.g. GABA and glutamatergic inhibition). Therefore, to further generalize our model to other species, we assessed whether it can also account for rats' behaviour in synchrony and temporal order judgments. Overall, we could tightly replicate rats' behaviour (rho = 0.981; see *Figure 2C-E*, *Figure 2—figure supplement 2*), including the effect of loudness on observed responses (*Figure 2D*). Interestingly, the unimodal temporal constants for rats were 4 times faster than for humans: such a different temporal tuning is reflected in higher sensitivity in rats for short lags (<0.1 s), and in humans for longer lags (*Figure 2C*). This fourfold difference in temporal tuning between rats and humans closely mirrors

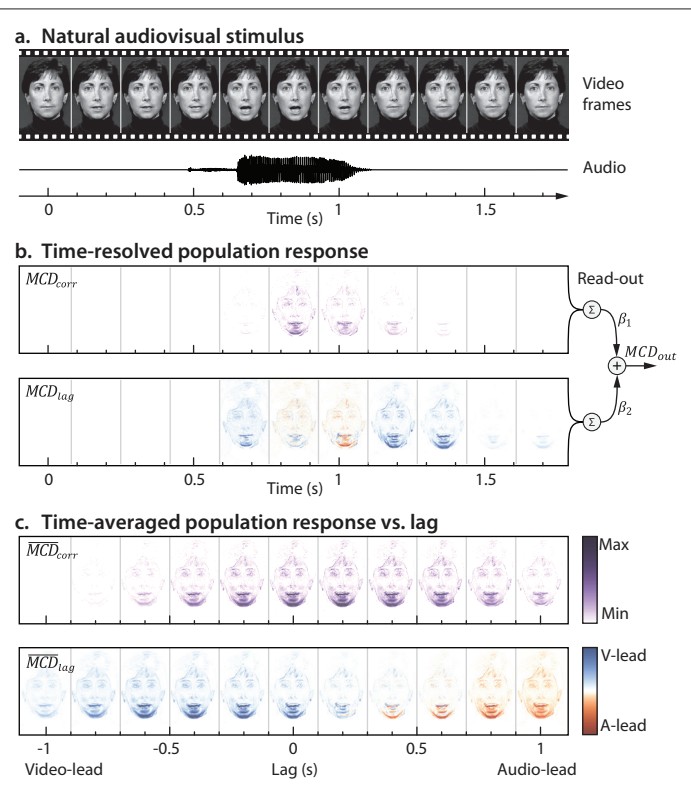

**Figure 3.** Ecological audiovisual stimuli and model responses. (**A**) displays the frames and soundtrack of a dynamic audiovisual stimulus over time (in this example, video and audio tracks are synchronous, and the actress utters the syllable /ta/). (**B**) shows how the dynamic population responses $MCD_{corr}$ and $MCD_{lag}$ vary across the frames of Panel A. Note how model responses highlight the pixels whose intensity changed with the soundtrack (i.e. the mouth area). The right side of Panel B represents the population read-out process, as implemented for the simulations in *Figure 2*: the population responses $MCD_{corr}$ and $MCD_{lag}$ are integrated over space (i.e. pixels) and time (i.e. frames), scaled and weighted by the gain parameters $\beta_{corr}$, and $\beta_{lag}$ and summed to obtain a single decision variable that is fed to the decision-making stage (see *Figure 1D*). (**C**) represents the time-averaged population responses $\overline{MCD_{corr}}$ and $\overline{MCD_{lag}}$ as a function of cross-modal lag (the central one corresponds to the time-averaged responses shown in **B**). Note how $\overline{MCD_{corr}}$ peaks at around zero lag and decreases with increasing lag (following the same trend shown in *Figure 1C*, left), while polarity of $\overline{MCD_{lag}}$ changes with the sign of the delay. The psychophysical data corresponding to the stimulus in this figure is shown in *Figure 2—figure supplement 1B*. See *Video 2* for a dynamic representation of the content of this figure.

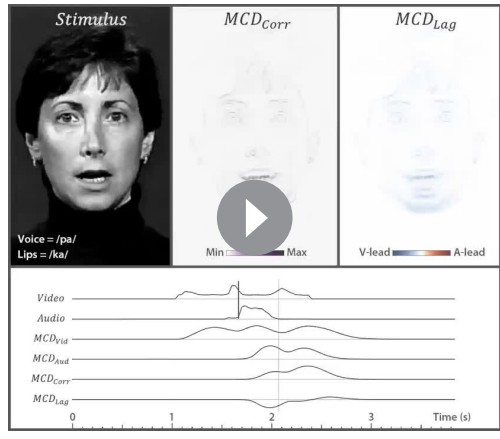

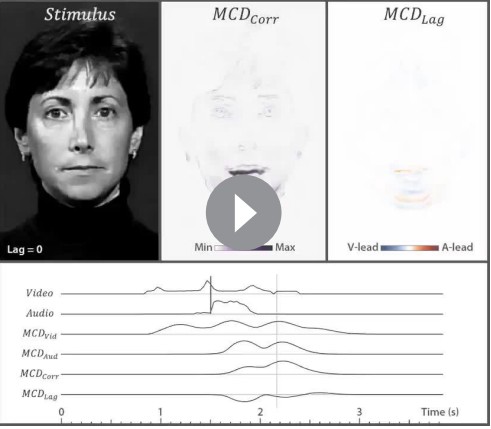

**Video 1.** The McGurk Illusion – integration of mismatching audiovisual speech. The soundtrack is from a recording where the actress utters the syllable /pa/, whereas in the video she utters /ka/. When the video and sound tracks are approximately synchronous, observers often experience the McGurk illusion, and perceive the syllable /ta/. To experience the illusion, try to recognize what the actress utters as we manipulate audiovisual lag. Note how the $MCD_{corr}$ population response clusters around the mouth area, and how its magnitude scales with the probability of experiencing the illusion. See **Video 2** for details.

https://elifesciences.org/articles/106122/figures#video1

**Video 2.** Population response to audiovisual speech stimuli. The top left panel displays the stimulus from **van Wassenhove et al., 2007**, where the actress utters the syllable /ta/. The central and right top panels represent the dynamic $MCD_{corr}(x, y, t)$ and $MCD_{lag}(x, y, t)$ population responses, respectively (**Equations 6 and 7**). The lower part of the video displays the temporal profile of the stimuli and model responses (averaged over space). The top two lines represent the stimuli: for the visual stimuli, the line represents the root-mean-squared difference of the pixel value from one frame to the next; the line for audio represents the envelope of the stimulus. $MCD_{Vid}$ and $MCD_{Aud}$ represents the output of the unimodal transient channels (averaged over space) that feed to the MCD (**Equation 2**). The two lower lines represent the $MCD_{corr}$ and $MCD_{lag}$ responses and correspond to the average of the responses displayed in the central and top-right panels This movie corresponds to the data displayed in **Figure 3A–B**. Note how the magnitude of $MCD_{corr}$ increases as the absolute lag decreases, while the polarity of $MCD_{lag}$ changes depending on which modality came first.

https://elifesciences.org/articles/106122/figures#video2

analogous interspecies differences in physiological rhythms, such as heart rate (~4.7 times faster in rats) and breathing rate (~6.3 times faster in rats) (**Agoston, 2017**).

While tuning the temporal constants of the model was necessary to account for the difference between humans and rats, this was not necessary to reproduce pharmacologically-induced changes in audiovisual time perception in rats (**Figure 2E**, **Figure 2—figure supplement 2F-G**), which could be accounted for solely by changes in the decision-making process(**Equation 10**). This suggests that the observed effects can be explained without altering low-level temporal processing. However, this does not imply that such changes did not occur—only that they were not required to reproduce the behavioural data in our simulations. Future studies using richer temporal stimuli—such as temporally modulated sequences that vary in frequency, rhythm, or phase—will be necessary to disentangle sensory and decisional contributions, as these stimuli can more selectively engage low-level temporal processing and better reveal whether perceptual changes arise from early encoding or later interpretive stages.

An asset of a low-level approach is that it allows one to inspect, at the level of individual pixels and frames, the features of the stimuli that determine the response of the model (i.e. the saliency maps). This is illustrated in **Figure 3** and **Videos 1 and 2** for the case of audiovisual speech, where model responses cluster mostly around the mouth area and (to a lesser extent) the eyes. These are the regions where pixels' luminance changes in synch with the audio track.

## Spatial determinants of audiovisual integration in humans and monkeys

Classic experiments on the spatial determinants of audiovisual integration usually require observers to localize the stimuli under systematic manipulations of the discrepancy and reliability (i.e. precision)

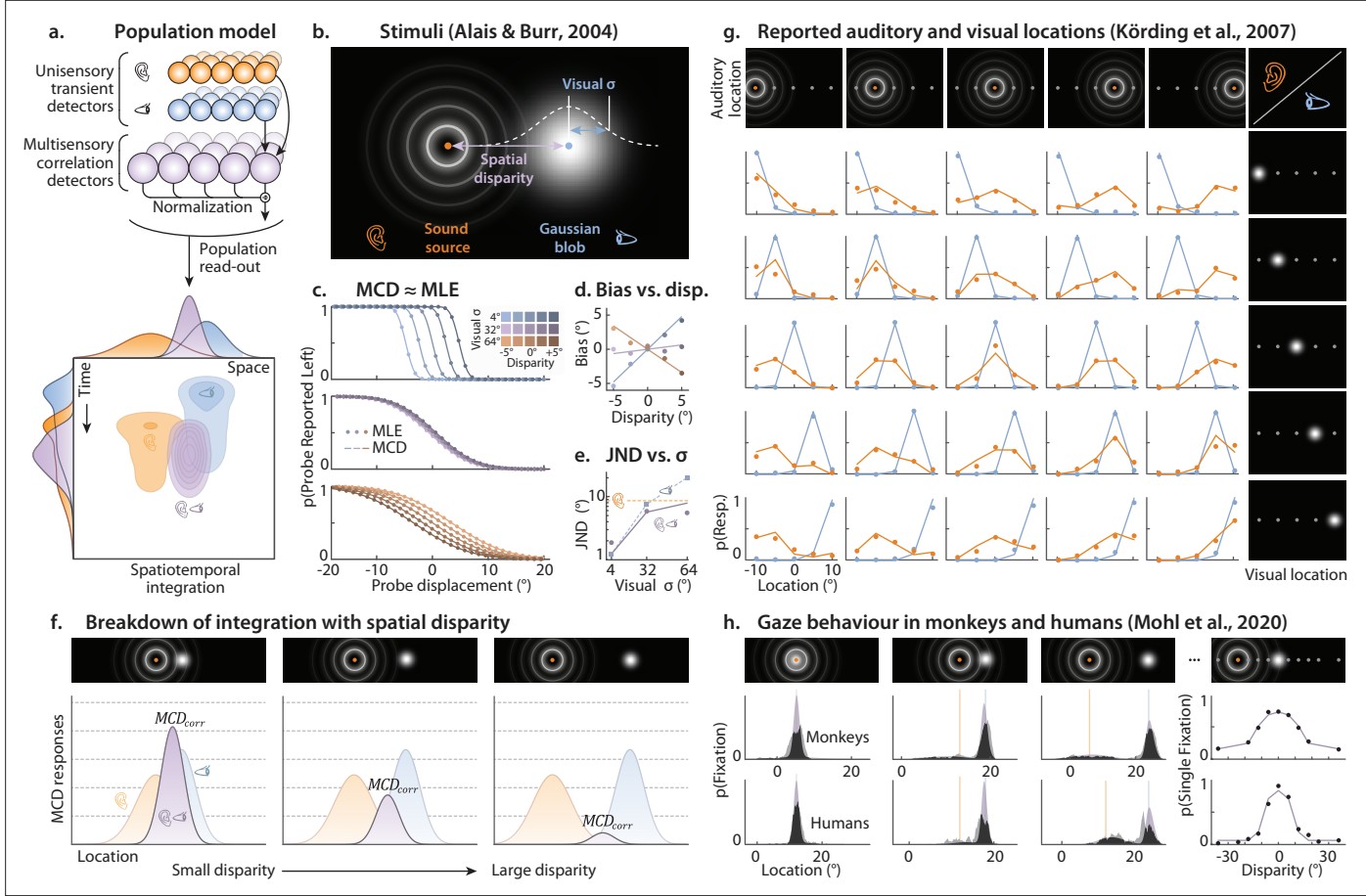

**Figure 4.** Audiovisual integration in space. (**A**) Top represents the Multisensory Correlation Detector (MCD) population model for spatialized audio. Visual and auditory input units receive input from corresponding spatial locations, and feed into spatially-tuned MCD units. The output of each MCD unit is eventually normalized by the total population output, so as to represent the probability distribution of stimulus location over space. The bottom part of Panel A represents the dynamic unimodal and bimodal population response over time and space (azimuth) and their marginals. When time is marginalized out, a population of MCDs implements integration as predicted by the maximum likelihood estimation (MLE) model. When space is marginalized, the output show the temporal response function of the model. In this example, visual and auditory stimuli were asynchronously presented from discrepant spatial locations (note how the blue and orange distributions are spatiotemporally offset). (**B**) shows a schematic representation of the stimuli used to test the MLE model by *Alais and Burr, 2004*. Stimuli were presented from different spatial positions, with a parametric manipulation of audiovisual spatial disparity and blob size (i.e. the standard deviation, σ of the blob). (**C**) shows how the bimodal psychometric functions predicted by the MCD (lines, see *Equation 16*) and the MLE (dots) models fully overlap. (**D**) shows how bimodal bias varies as a function of disparity and visual reliability (see legend on the left). The dots correspond to the empirical data from participant LM, while the lines are the predictions of the MCD model (compare to *Figure 2A*, of Alais and Burr, 2004). (**E**) shows how the just noticeable differences (just noticeable difference JND, the random localization error) vary as a function of blob size. The blue squares represent the visual JNDs, the purple dots the bimodal JNDs, while the dashed orange line represents the auditory JND. The continuous line shows the JNDs predicted by the MCD population model (compare to *Figure 2B*, of *Alais and Burr, 2004*). (**F**) represents the breakdown of integration with spatial disparity. The magnitude of the MCD population output (*Equation 8*, shown as the area under the curve of the bimodal response) decreases with increasing spatial disparity across the senses. This can be then transformed into a probability for a common cause (*Equation 19*). (**G**) represents the stimuli and results of the experiment used by *Körding et al., 2007* to test the Bayesian Causal Inference (BCI) model. Auditory and visual stimuli originate from one of five spatial locations, spanning a range of 20° . The plots show the perceived locations of visual (blue) and auditory (orange) stimuli for each combination of audiovisual spatial locations. The dots represent human data, while the lines represent the responses of the MCD population model. (**H**) shows the stimuli and results of the experiment of *Mohl et al., 2020*. The plots on the right display the probability of a single (vs. double) fixation (top monkeys, bottom humans). The dots represent human data, while the lines represent the responses of the MCD population model. The remaining panels show the histogram of the fixated locations in bimodal trials: the jagged histograms are the empirical data, while the smooth ones are the model prediction (zero free parameters). The regions of overlap between empirical and predicted histograms are shown in black.

The online version of this article includes the following figure supplement(s) for figure 4:

**Figure supplement 1.** Multisensory Correlation Detector (MCD) model for trimodal integration.

of the spatial cues (*Alais and Burr, 2004*; *Figure 4*). This allows one to assess how unimodal cues are weighted and combined to give rise to phenomena such as the ventriloquist illusion (*Stratton, 1897*). When the spatial discrepancy across the senses is low, observers' behaviour is well described by Maximum Likelihood Estimation (MLE; *Alais and Burr, 2004*), where unimodal information is combined in a statistically optimal fashion, so as to maximize the precision (reliability) of the multi-modal percept (see *Equations 11–14*, Methods).

Given that both the MLE and the MCD operate by multiplying unimodal inputs (see Methods), the time-averaged MCD population response (*Equation 16*) is equivalent to MLE (*Figure 4A*). This can be illustrated by simulating the experiment of *Alais and Burr, 2004* using both models. In this experiment, observers had to report whether a probe audiovisual stimulus appeared left or right of a standard. To assess the weighing behaviour resulting from multisensory integration, they manipulated the spatial reliability of the visual stimuli and the disparity between the senses (*Figure 4B*). *Figure 4C* shows that the integrated percept predicted by the two models is statistically indistinguishable. As such, a population of MCDs (*Equation 16*) can jointly account for the observed bias and precision of the bimodal percept (*Figure 4D–E*), with zero parameters. A MATLAB implementation of this simulation is included as *Source code 1*.

While fusing audiovisual cues is a sensible solution in the presence of minor spatial discrepancies across the senses, integration eventually breaks down with increasing disparity (*Chen and Vroomen, 2013*)—when the spatial (or temporal) conflict is too large, visual and auditory signals may well be unrelated. To account for the breakdown of multisensory integration in the presence of intersensory conflicts, Körding and colleagues proposed the influential Bayesian Causal Inference (BCI) model (*Körding et al., 2007*), where uni- and bimodal location estimates are weighted based on the probability that the two modalities share a common cause (*Equation 17*). The BCI model was originally tested in an experiment in which sound and light were simultaneously presented from one of five random locations, and observers had to report the position of both modalities (*Körding et al., 2007*; *Figure 4G*). Results demonstrate that visual and auditory stimuli preferentially bias each other when the discrepancy is low, with the bias progressively declining as the discrepancy increases.

Also a population of MCDs can compute the probability that auditory and visual stimuli share a common cause (*Figure 1B and D*; *Figure 4F*, *Equation 19*), therefore, we can test whether it can also implement BCI. For that, we simulated the experiment of Körding and colleagues, and fed the stimuli to a population of MCDs (*Equations 18-20*) which near-perfectly replicated the empirical data (rho = 0.99)–even slightly outperforming the BCI model. A MATLAB implementation of this simulation is included as *Source code 1*.

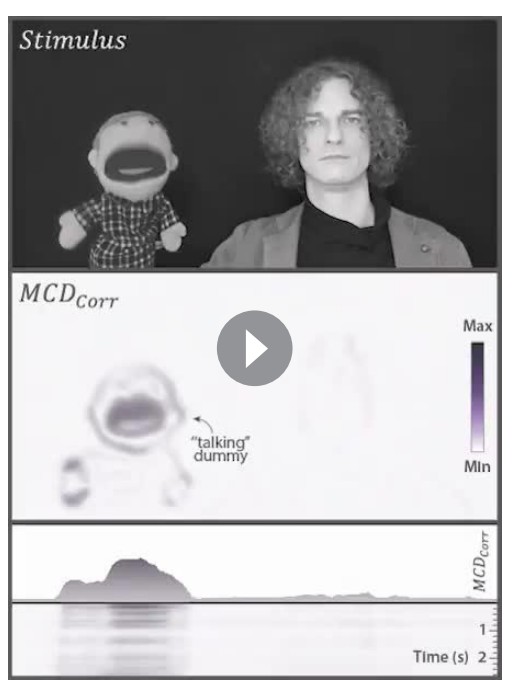

**Video 3.** The ventriloquist illusion. The top panel represents a video of a performing ventriloquist. The voice of the dummy was edited (pitch-shifted) and added in post-production. The second panel represents the dynamic $MCD_{corr}(x, y, t)$ population response to a blurred version of the video (*Equation 6*). The third panel shows the distribution of population responses along the horizontal axis (obtained by averaging the upper panel over the vertical dimension). This represents the dynamic, real-life version of the bimodal population response shown in *Figure 4A* for the case of minimalistic audiovisual stimuli. The lower panel represents the same information as the panel above displayed as a rolling timeline. For this video, the population response was temporally aligned to the stimuli to compensate for lags introduced by the temporal filters of the model. Note how the population response spatially follows the active speaker, hence capturing the sensed location of the audiovisual event towards correlated visuals.

https://elifesciences.org/articles/106122/figures#video3

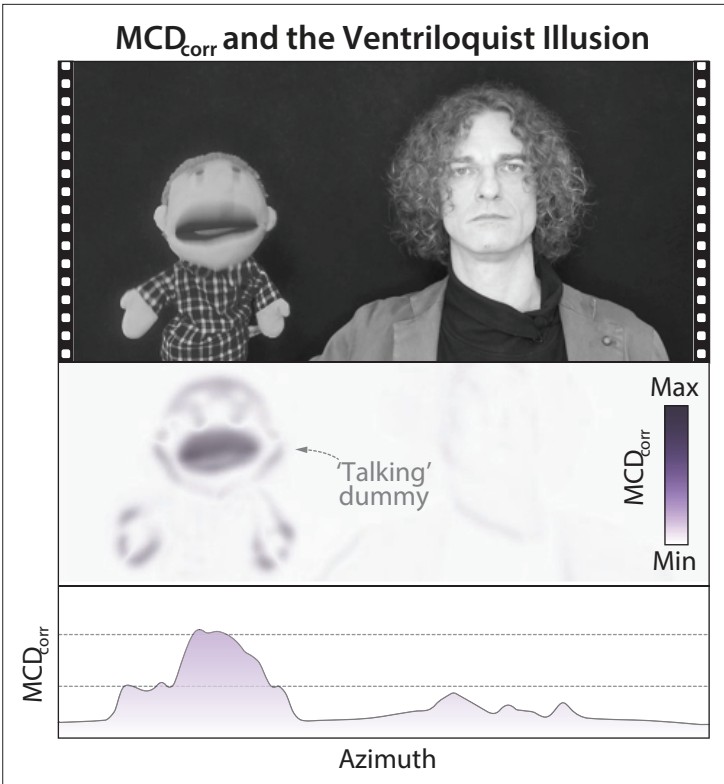

**Figure 5.** Multisensory Correlation Detector (MCD) and the Ventriloquist Illusion. The upper panel represents still frame of a performing ventriloquist. The central panel represents the MCD population response. The lower plot represents the horizontal profile of the MCD response for the same frame. Note how the population response clusters on the location of the dummy, where more pixels are temporally correlated with the soundtrack.

To test the generalizability of these findings across species and behavioural paradigms, we simulated an experiment in which monkeys (*Macaca mulatta*) and humans directed their gaze toward audiovisual stimuli presented at varying spatial disparities (*Figure 4H*; *Mohl et al., 2020*). If observers infer a common cause, they tend to make a single fixation; otherwise, two—one for each modality. As expected, the probability of a single fixation decreased with increasing disparity (*Figure 4H*, right). This pattern was captured by a population of MCDs: $MCD_{corr}$ values were used to fit the probability of single vs. double saccades as a function of disparity (*Equation 21*, *Figure 4H*, right). Critically, using this fit, the model was then able to predict the full distribution of gaze locations (*Equation 20*, *Figure 4H*, left) in both species with zero additional free parameters. A Matlab implementation of this simulation is included as *Source code 1*.

Taken together, these simulations show that behaviour consistent with BCI and MLE naturally emerges from a population of MCDs. Unlike BCI and MLE, however, the MCD population model is both image- and sound-computable, and it explicitly represents the spatiotemporal dynamics of the process (*Figure 4A*, bottom; *Figure 3B*; *Figure 5*; *Figure 6B–C*). On one hand, this enables the model to be applied to complex, dynamic audiovisual stimuli—such as real-life videos—that were previously *off limits* to traditional BCI and MLE frameworks, whose probabilistic, non–stimulus-computable formulations prevent them from operating directly on such inputs. On the other, it permits direct, time-resolved comparisons between model responses and neurophysiological measures (*Pesnot Lerousseau et al., 2022*).

As a practical demonstration, we applied the model (*Equation 6*) to a real-life video of a performing ventriloquist (*Figure 5*). The population response dynamically tracked the active talker, clustering around the dummy's face whenever it produced speech *Video 3*.

## Spatial orienting and audiovisual saliency maps

Multisensory stimuli are typically salient, and a vast body of literature demonstrates that spatial attention is commonly attracted to audiovisual stimuli (*Talsma et al., 2010*). This aspect of multi-sensory perception is naturally captured by a population of MCDs, whose dynamic response explic-itly represents the regions in space with the highest audiovisual correspondence for each point in time. Therefore, for a population of MCDs to provide a plausible account for audiovisual integration, such dynamic saliency maps should be able to predict human audiovisual gaze behaviour, in a purely bottom-up fashion and with no free parameters. *Figure 6A* shows the stimuli and eye-tracking data from the experiment of *Coutrot and Guyader, 2015*, in which observers passively watched a video of four persons talking. Panel B shows the same eye-tracking data plotted over the corresponding MCD population response: across 20 observers, and 15 videos (for a total of over 16,000 frames), gaze was on average directed towards the locations (i.e. pixels) yielding the top 2% MCD response (*Figure 6D*, *Equations 22; 23*). The tight correspondence of predicted and empirical salience is illustrated in *Figure 6C* and *Video 4*: note how population responses peak based on the active speaker.

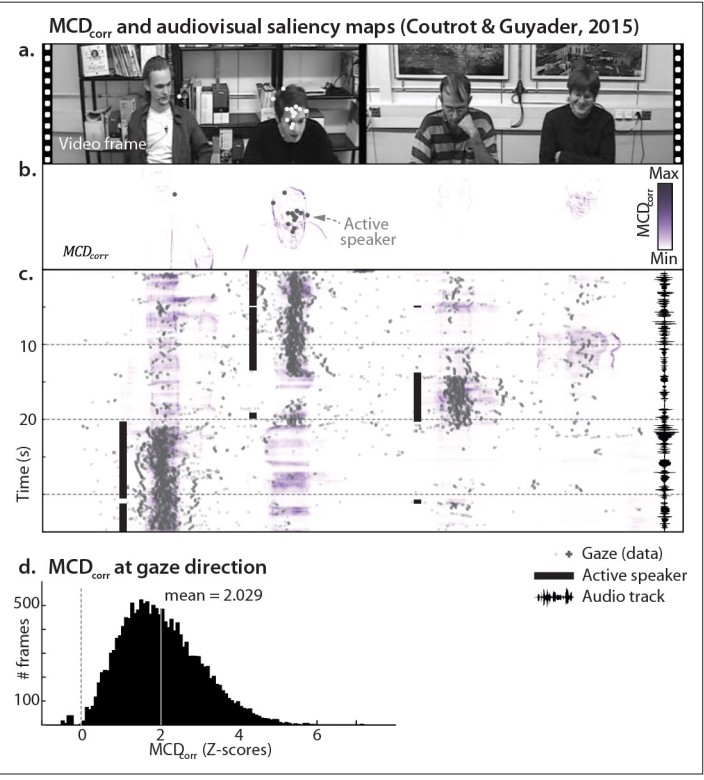

**Figure 6.** Audiovisual saliency maps. (**A**) represents a still frame of *Coutrot and Guyader, 2015* stimuli. The white dots represent gaze direction of the various observers. (**B**) represents the Multisensory Correlation Detector (MCD) population response for the frame in Panel A. The dots represent observed gaze direction (and correspond to the white dots of Panel A). (**C**) represents how the MCD response varies over time and azimuth (with elevation marginalized-out). The black solid lines represent the active speaker, while the waveform on the right displays the soundtrack. Note how the MCD response was higher for the active speaker. (**D**) shows the distribution of model response at gaze direction (see Panel B) across all frames and observers in the database. Model response was normalized for each frame (Z-scores). The y axis represents the number of frames. The vertical gray line represents the mean. See *Video 4* for a dynamic representation of the content of this figure.

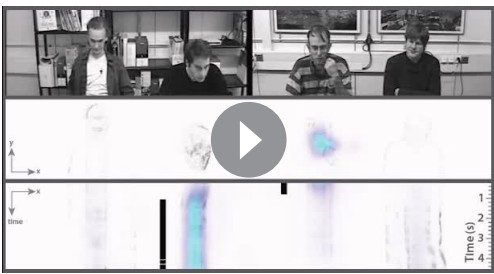

**Video 4.** Audiovisual saliency maps. The top panel represents Movie 1 from *Coutrot and Guyader, 2015*. The central panel represents $MCD_{corr}$ in gray scales, while the colorful blobs represent observers' gaze direction during passive viewing. The lower panel represents how the $MCD_{corr}$ and gaze direction (co)vary over time and azimuth (with elevation marginalized-out). The black solid lines represent the active speaker. For the present simulations, movies were converted to grayscale and the upper and lower sections of the videos (which were mostly static) were cropped. Note how gaze is consistently directed towards the regions of the frames displaying the highest audiovisual correlation.

https://elifesciences.org/articles/106122/figures#video4

## Discussion

This study demonstrates that elementary audiovisual analyses are sufficient to replicate behaviours consistent with multisensory perception in mammals. The proposed image- and sound-computable model, composed of a population of biologically plausible elementary processing units, provides a stimulus-driven framework for multisensory perception that transforms raw audiovisual input into behavioural predictions. Starting directly from pixels and audio samples, our model closely matched observed behaviour across a wide range of phenomena—including multisensory illusions, spatial orienting, and causal inference—with average correlations above 0.97. This was tested in a large-scale simulation spanning 69 audiovisual experiments, seven behavioural tasks, and data from 534 humans, 110 rats, and two monkeys.

We define a stimulus-computable model as one that receives input directly from the stimulus—such as raw images and sound waveforms—rather than from abstracted descriptors like lag, disparity, or reliability. Framed in Marr's terms, stimulus-computable models operate at the algorithmic level, specifying how sensory information is represented and processed. This contrasts with computational-level models, such as Bayesian ideal observers, which define the goals of perception (e.g. maximizing reliability *Alais and Burr, 2004*; *Ernst and Banks, 2002*) without specifying how those goals are achieved. Rather than competing with such normative accounts, the MCD provides a mechanistic substrate that could plausibly implement them. By operating directly on realistic audiovisual signals, our population model captures the richness of natural sensory input and directly addresses the problem of how biological systems represent and process multisensory information (*Burge, 2020*). This allows the MCD to generate precise, stimulus-specific predictions across tasks, including subtle differences in behavioural outcomes that arise from the structure of individual stimuli (see *Figure 2—figure supplement 1K*).

The present approach naturally lends itself to be generalized and tested against a broad range of tasks, stimuli, and responses—as reflected by the breadth of the experiments simulated here. Among the perceptual effects emerging from elementary signal processing, one notable example is the scaling of subjective audiovisual synchrony with sound source distance (*Alais and Carlile, 2005*). As sound travels slower than light, humans compensate for audio delays by adjusting subjective synchrony based on the source's distance scaled by the speed of sound. Although this phenomenon appears to rely on explicit physics modelling, our simulations demonstrate that auditory cues embedded in the envelope (*Figure 2B*, left) are sufficient to scale subjective audiovisual synchrony. In a similar fashion, our simulations show that phenomena such as the McGurk illusion, the subjective timing of natural audiovisual stimuli, and saliency detection may emerge from elementary operations performed at pixel level, bypassing the need for more sophisticated analyses such as image segmentation, lip or face tracking, 3D reconstruction, etc (*Chandrasekaran et al., 2009*). Elementary, general-purpose operations on natural stimuli can drive complex behaviour, sometimes even in the absence of advanced perceptual and cognitive contributions. Indeed, it is intriguing that a population of MCDs, a computational architecture originally proposed for motion vision in insects, can predict speech illusions in humans.

The fact that identical low-level analyses can account for all of the 69 experiments simulated here directly addresses several open questions in multisensory research. For instance, psychometric functions for speech and non-speech stimuli often differ significantly (*Vatakis et al., 2008*). This has been interpreted as evidence that speech may be special and processed via dedicated mechanisms

(*Tuomainen et al., 2005*). However, identical low-level analyses are sufficient to account for all observed responses, regardless of the stimulus type (*Figure 2*, *Figure 2—figure supplements 1 and 2*). This suggests that most of the differences in psychometric curves across classes of stimuli (e.g. speech vs. non-speech vs. clicks-&-flashes) are due to the low-level features of the stimuli themselves, not how the brain processes them. Similarly, experience and expertise also modulate multisensory perception. For example, audiovisual simultaneity judgments differ significantly between musicians and non-musicians (*Lee and Noppeney, 2011*) (see *Figure 2—figure supplement 1C*). Likewise, the McGurk illusion (*Freeman et al., 2013*) and subjective audiovisual timing (*Petrini et al., 2009*) vary over the lifespan in humans, and following pharmacological interventions in rats (*Al Youzbaki et al., 2023*; *Schormans and Allman, 2023*) (see *Figure 2—figure supplement 1E and J* and *Figure 2—figure supplement 2F-G*). Our simulations show that adjustments at the decision-making level are sufficient to account for these effects, without requiring structural or parametric changes to low-level perceptual processing across observers or conditions.

Although the same model explains responses to multisensory stimuli in humans, rats, and monkeys, the temporal constants vary across species. For example, the model for rats is tuned to temporal frequencies over four times higher than those for humans. This not only explains the differential sensitivity of humans and rats to long and short audiovisual lags, but it also mirrors analogous inter-species differences in physiological rhythms, such as heart and breathing rates (*Agoston, 2017*). Previous research has shown that physiological arousal modulates perceptual rhythms within individuals (*Legrand et al., 2018*). It is an open question whether the same association between multisensory temporal tuning and physiological rhythms persists in other mammalian systems. Conversely, no major differences in the model's spatial tuning were found between humans and macaques, possibly reflecting the close phylogenetic link between the two species.

How might these computations be implemented neurally? In a recent study (*Pesnot Lerousseau et al., 2022*), we identified neural responses in the posterior superior temporal sulcus, superior temporal gyrus, and left superior parietal gyrus that tracked the output of an MCD model during audiovisual temporal tasks. Participants were presented with random sequences of clicks and flashes while performing either a causality judgment or a temporal order judgment task. By applying a time-resolved encoding model to MEG data, we demonstrated that MCD dynamics aligned closely with stimulus-evoked cortical activity. The present study considerably extends the scope of the MCD framework, allowing it to process more naturalistic stimuli and to account for a broader range of behaviours—including cue combination, attentional orienting, and gaze-based decisions. This expansion opens the door to new neurophysiological investigations into the implementation of multisensory integration. For instance, the dynamic, spatially distributed population responses generated by the MCD (see videos) can be directly compared with neural population activity recorded using techniques such as ECoG, Neuropixels, or high-density fMRI—similar to previous efforts that linked the Bayesian Causal Inference model to neural responses during audiovisual spatial integration (*Rohe et al., 2019*; *Aller and Noppeney, 2019*; *Rohe and Noppeney, 2015*). Such comparisons may help bridge algorithmic and implementational levels of analysis, offering concrete hypotheses about how audiovisual correspondence detection and integration are instantiated in the brain.

An informative outcome of our simulations is the model's ability to predict spontaneous gaze direction in response to naturalistic audiovisual stimuli. Saliency, the property by which some elements in a display stand out and attract observer's attention and gaze direction, is a popular concept in both cognitive and computer sciences (*Itti et al., 1998*). In computer vision, saliency models are usually complex and rely on advanced signal processing and semantic knowledge—typically with tens of millions of parameters (*Chen et al., 2023*; *Coutrot, 2025*). Despite successfully predicting gaze behaviour, current audiovisual saliency models are often computationally expensive, and the resulting maps are hard to interpret and inevitably affected by the datasets used for training (*Adebayo et al., 2023*). In contrast, our model detects saliency 'out of the box,' without any free parameters, and operating purely at the individual pixel level. The elementary nature of the operations performed by a population of MCDs returns saliency maps that are easy to interpret: salient points are those with high audiovisual correlation. By grounding multisensory integration and saliency detection in biologically plausible computations, our study offers a new tool for machine perception and robotics to handle multimodal inputs in a more human-like way, while also improving system accountability.

This framework also provides a solution for self-supervised and unsupervised audiovisual learning in multimodal machine perception. A key challenge when handling raw audiovisual data is solving the causal inference problem—determining whether signals from different modalities are causally related or not (*Körding et al., 2007*). Models in machine perception often depend on large, labelled data-sets for training. In this context, a biomimetic module that handles saliency maps, audiovisual corre-spondence detection, and multimodal fusion can drive self-supervised learning through simulated observers, thereby reducing the dependency on labelled data (*Shahabaz and Sarkar, 2024*; *Arand-jelovic and Zisserman, 2017*; *Ngiam et al., 2011*). Furthermore, the simplicity of our population-based model provides a computationally efficient alternative for real-time multisensory integration in applications such as robotics, AR/VR, and other low-latency systems.

Although a population of MCDs can explain when phenomena such as the McGurk Illusion occur, it does not explain the process of phoneme categorization that ultimately determines what syllable is perceived (*Magnotti and Beauchamp, 2017*). More generally, it is well known that cognitive and affective factors modulate our responses to multisensory stimuli (*Stein, 2012*). In particular, the model does not currently incorporate linguistic mechanisms or top-down predictive processes, which play a central role in audiovisual speech perception—such as the integration of complementary articula-tory features, lexical expectations, or syntactic constraints (*Campbell, 2008*; *Peelle and Sommers, 2015*; *Summerfield, 1987*; *Tye-Murray et al., 2007*). While a purely low-level model does not directly address these issues, the modularity of our approach makes it possible to extend the system to include high-level perceptual, cognitive, and affective factors. What is more, although this study focused on audiovisual integration in mammals, the same approach can be naturally extended to other instances of sensory integration (e.g. visuo- and audio-tactile) and animal classes. A possible extension of the model for trimodal integration is included in *Figure 4—figure supplement 1*.

Besides simulating behavioural responses, a stimulus-computable approach necessarily makes explicit all the intermediate steps of sensory information processing. This opens the system to inspection at all of its levels, thereby allowing for direct comparisons with neurophysiology (*Pesnot Lerousseau et al., 2022*). In insect motion vision, this transparency made it possible for the Hassenstein-Reichardt detector to act as a searchlight to link computation, behavior, and physiology at the scale of individual cells (*Serbe et al., 2016*). Being based on formally identical computational principles (*Parise and Ernst, 2016*), the present approach holds the same potential for multisensory perception.

## Methods
### The MCD population model

The architecture of each MCD unit used here is the same as described in *Parise and Ernst, 2025*, however, units here receive time-varying visual and auditory input from spatiotopic receptive fields. The input stimuli ($s$) consist of luminance level and sound amplitude varying over space and time and are denoted as $s_m(x, y, t)$ – with $x$ and $y$, representing the spatial coordinates along the horizontal and vertical axes, $t$ is the temporal coordinate, and $m$ is the modality (video and audio). When the input stimulus is a movie with mono audio, the visual input to each unit is a signal representing the luminance of a single pixel over time, while the auditory input is the amplitude envelope (later, we will consider more complex scenarios where auditory stimuli are also spatialized).

Each unit operates independently and detects changes in unimodal signals over time by temporal filtering based on two biphasic impulse response functions that are 90° out of phase (i.e. a quadrature pair). A physiologically plausible implementation of this process has been proposed by *Adelson and Bergen, 1985* and consists of linear filters of the form:

$$f_n(t) = \left(\frac{t}{\tau_{bp}}\right)^n \cdot e^{-\frac{t}{\tau_{bp}}} \cdot \left[\frac{1}{n!} - \frac{1}{(n+2)!} \cdot \left(\frac{t}{\tau_{bp}}\right)^2\right] \tag{1}$$

The phase of the filter is determined by $n$, which based on *Emerson et al., 1992* takes the values of 6 for the fast filter and 9 for the slow one. The temporal constant of the filters is deter-mined by the parameter $\tau_{bp}$; in humans, its best fitting value is 0.045 s for vision and 0.0367 s for audition. In rats, the fitted temporal constant for vision and audition are nearly identical and their value is 0.010 s.

Fast and slow filters are applied to each unimodal input signal and the two resulting signals are squared and then summed. After that, a compressive non-linearity (square-root) is applied to the output, so as to constrain it within a reasonable range (*Adelson and Bergen, 1985*). Therefore, the output of each unimodal unit feeding into the correlation detector takes the following form.

$$MCD_{mod}(x, y, t) = \sqrt{\left[s_m(x, y, t) * f_6(t)\right]^2 + \left[s_m(x, y, t) * f_9(t)\right]^2},$$ (2)

where $mod = vid, aud$ represents the sensory modality and $*$ is the convolution operator.

As in the original version (*Parise and Ernst, 2016*), each MCD consists of two sub-units, in which the unimodal input is low-pass filtered and multiplied as follows.

$$u_1(x, y, t) = MCD_{vid}(x, y, t) \cdot \left[MCD_{aud}(x, y, t) * f_{lp}(t)\right]$$ (3)

$$u_2(x, y, t) = MCD_{aud}(x, y, t) \cdot \left[MCD_{vid}(x, y, t) * f_{lp}(t)\right]$$ (4)

The impulse response of the low-pass filter of each sub-unit takes the form.

$$f_{lp}(t) = \frac{t}{\tau_{lp}} \cdot e^{-\frac{t}{\tau_{lp}}},$$ (5)

where $\tau_{lp}$ represents the temporal constant, and its estimated value is 0.180 s for humans and 0.138 s for rats.

The response of the sub-units is eventually multiplied to obtain $MCD_{corr}$, which represents the local spatiotemporal audiovisual correlation, and subtracted to obtain $MCD_{lag}$ which describes the relative temporal order of vision and audition

$$MCD_{corr}(x, y, t) = u_1(x, y, t) \cdot u_2(x, y, t)$$ (6)

$$MCD_{lag}(x, y, t) = u_1(x, y, t) - u_2(x, y, t)$$ (7)

The outputs $MCD_{corr}(x, y, t)$ and $MCD_{lag}(x, y, t)$ are the final products of the MCD.

The **temporal constants** of the filters were fitted using the Bayesian Adaptive Direct Search (BADS) algorithm (*Acerbi and Ma, 2017*), set to maximize the correlation between the empirical and predicted psychometric functions of the temporal determinants of multisensory integration. For humans, that included all studies in *Figure 2—figure supplement 1*, besides Patient PH; For rats, it included all non-pharmacological studies in *Figure 2—figure supplement 2*. To minimize the effect of starting parameters, the fitting was performed 200 times using random starting values (from 0.001 to 1.5 s). The parameters estimated using BADS were further refined using the *fminsearch* algorithm in MATLAB. Parameter estimation was considerably compute-intensive, hence, the amount of data had to be reduced by rescaling the videos to 15% of the original size (without affecting the frame rate). Besides reducing run-time, this simulates the (Gaussian) spatial pooling occurring in early visual pathways. Details on parameter fitting are provided below.

Now that the MCD population is defined and its parameters are fully constrained, what remains to be explained is how to read out, from the dynamic population responses, the relevant information that is needed to generate a behavioral response, such as eye movements, button presses, nose, or lick contacts, etc. While the MCD units are task-independent and operate in a purely bottom-up fashion, the exact nature of the read-out and decision-making processes depends on the behavioral task, which ultimately determines how to weigh and combine the dynamic population responses. Given that in the present study, we consider different types of behavioral experiments (investigating audiovisual integration through temporal tasks, spatial tasks, and passive observation), the read-out process for each task will be described in separate sections.

## Modeling the temporal determinants of multisensory integration

The experiments on the temporal constraints of audiovisual integration considered here rely on psychophysical forced-choice tasks to assess the effects of cross-modal lags on perceived synchrony,

temporal order, and the McGurk illusion. In humans, such experiments entail pressing one of two buttons; in rats, nose-poking or licking one of two spouts. In the case of simultaneity judgments, on each trial, observers reported whether visual and auditory stimuli appeared synchronous or not ($resp = \{yes, no\}$). In temporal order judgments, observers reported which modality came first ($resp = \{vision\,first,\,audition\,first\}$). Finally, in the case of the McGurk, observers (humans) had to report what syllable they heard (e.g. 'da,' 'ga,' or 'ba,' usually recoded as 'fused' and 'non-fused percept' or 'illusion present' and 'illusion absent'). When plotted against lag, the resulting empirical psychometric functions describe how audiovisual timing affects perceived synchrony, temporal order, and the McGurk illusion. Here, we model these tasks using the same read-out and decision-making process.

To account for observed responses, the dynamic, high-bandwidth population responses must be transformed (compressed) into a single number, representing response probability for each lag. In line with standard procedures (**Parise and Ernst, 2017**), this was achieved by integrating $MCD_{corr}(x, y, t)$ and $MCD_{lag}(x, y, t)$ over time and space, so as to obtain two summary decision variables

$$\overline{MCD_{corr}} = \sum_x \sum_y \sum_t MCD_{corr}(x, y, t) \tag{8}$$

and

$$\overline{MCD_{lag}} = \sum_x \sum_y \sum_t MCD_{lag}(x, y, t) \tag{9}$$

The temporal window for these analyses consisted of the duration of each stimulus, plus 2 s before and after (during which the audio was silent, and the video displayed a still frame); the exact extension of the temporal window used for the analyses had minimal effect on the results. $\overline{MCD_{corr}}$ and $\overline{MCD_{lag}}$ are eventually linearly weighted and transformed into response probabilities through a cumulative normal function as follows

$$p_{MCD}(resp) = \Phi\left(\beta_{crit} + \beta_{corr} \cdot \overline{MCD_{corr}} + \beta_{lag} \cdot \overline{MCD_{lag}}\right) \tag{10}$$

Here, $\Phi$ represents the cumulative normal, $\beta_{corr}$ and $\beta_{lag}$ are linear coefficients that weigh and scale $\overline{MCD_{corr}}$ and $\overline{MCD_{lag}}$. $\beta_{crit}$ is a bias term, which corresponds to the response criterion. Finally, $p_{MCD}(resp)$ is the probability of a response, which is $p_{MCD}(Synchronous)$ for the simultaneity judgment task, $p_{MCD}(Audio\,first)$ for the temporal order judgment task, and $p_{MCD}(Fusion)$ for the McGurk task. Note how **Equation 9** entails that simultaneity judgments, temporal order judgments, and the McGurk illusions are all simulated using the very same model architecture.

With the population model fully constrained (see above), the only free parameters in the present simulations are the ones controlling the decision-making process: $\beta_{crit}$, $\beta_{corr}$, and $\beta_{lag}$. These were separately fitted for each experiment using *fitglm* MATLAB (binomial distribution and probit link function). For the simulations of the effects of distance on audiovisual temporal order judgments in humans (**Alais and Carlile, 2005**; **Figure 2B**), and the manipulation of loudness in rats (**Schormans and Allman, 2018**; **Figure 2D**) $\beta_{crit}$, $\beta_{corr}$, and $\beta_{lag}$ were constant across conditions (i.e. distance **Alais and Carlile, 2005** or loudness **Schormans and Allman, 2018**). This way, differences in the psychometric functions as a function of distance (**Alais and Carlile, 2005**) or loudness (**Schormans and Allman, 2018**) of the stimuli are fully explained by the MCD. Overall, the model provided a good fit to the empirical psychometric functions, and the average Pearson correlation between human and model response (weighted by sample size) is 0.981 for humans and 0.994 for rats. Naturally, model-data correlation varied across experiments, largely due to sample size. This can be appreciated when the MCD-data correlation for each experiment is plotted against the number of trials for each lag in a funnel plot (see **Figure 2—figure supplement 1I**). The number of trials for each lag determines the binomial error for data points in **Figure 2—figure supplements 1 and 2**; accordingly, the funnel plot shows that lower MCD-data correlation is more commonly observed for curves based on smaller sample size. This shows that the upper limit in the MCD-data correlation is mostly constrained by the reliability of the dataset, rather than systematic errors in the model.

To assess the contribution of the low-level properties of the stimuli on model's performance, we ran a permutation test, where psychometric curves were generated (**Equation 10**) using stimuli from different experiments (but with the same manipulation of lag). If the low-level properties of the stimuli

play a significant role, the correlation between data and model with permuted stimuli should be lower than with non-permuted stimuli. For this permutation, we used the data from simultaneity judgment tasks on humans (*Figure 2—figure supplement 1*), as with 27 individual experiments, there are enough permutations to render the test meaningful. For that, we used the temporal constants of the MCD fitted before, so that each psychometric curve, each permutation had three free parameters ($\beta_{crit}$, $\beta_{corr}$, and $\beta_{lag}$). The results from 200 k permutations demonstrate that the goodness of fit obtained with the original stimuli is superior to that of permuted stimuli. Specifically, the permuted distribution of the mean Pearson correlation of predicted vs. empirical psychometric curves had a mean of 0.972 ($\sigma$=0.0036), while such a correlation rose to 0.989 when the MCD received the original stimuli.

## Modeling the spatial determinants of multisensory integration

Most studies on audiovisual space perception only investigated the horizontal spatial dimension; hence the stimuli can be reduced to $s_m(x,t)$ instead of $s_m(x,y,t)$, as in the simulations above (see *Equation 2*). Additionally, based on preliminary observations, the output $MCD_{lag}$ does not seem necessary to account for audiovisual integration in space, hence, only the population response $MCD_{corr}(x,t)$ (see *Equation 6*) will be considered here.

*MCD and MLE – simulation of Alais and Burr, 2004*. In its general form, the MLE model can be expressed probabilistically as $p_{MLE}(x) \propto p_{vid}(x) \cdot p_{aud}(x)$, where $p_{vid}(x)$ and $p_{aud}(x)$ represent the probability distribution of the unimodal location estimate (i.e. likelihood functions), and $p_{MLE}(x)$ is the bimodal distribution. When the $p_{vid}(x)$ and $p_{aud}(x)$ follow a Gaussian distribution, also $p_{MLE}(x)$ is Gaussian, with the variance ($\sigma^2_{MLE}$) equal to the product divided by the sum of the unimodal variances:

$$\sigma^2_{MLE} = \frac{\sigma^2_{vid} \cdot \sigma^2_{aud}}{\sigma^2_{vid} + \sigma^2_{aud}} \tag{11}$$

and the mean consisting of a weighted average of the unimodal means

$$\mu_{MLE} = \omega_{vid} \cdot \mu_{vid} + \omega_{aud} \cdot \mu_{aud} \tag{12}$$

where

$$\omega_{vid} = \frac{1/\sigma^2_{vid}}{1/\sigma^2_{vid} + 1/\sigma^2_{aud}} \tag{13}$$

and

$$\omega_{aud} = 1 - \omega_{vid} \tag{14}$$

To test whether a population of MCDs can replicate the study of *Alais and Burr, 2004*, this simulation has two main goals. The first one is to compare the predictions of the MCD with that of the MLE model; the second one is to test whether the MCD model can predict observers' responses.

To simulate the study of *Alais and Burr, 2004*, we first need to generate the stimuli. The visual stimuli consisted of a 1-D Gaussian luminance profiles, presented for 10ms. Their standard deviations, which determined visual spatial reliability, were defined as the standard deviation of the visual psychometric functions. Likewise, the auditory stimuli also consisted of a 1-D Gaussian sound intensity profile (with a standard deviation determined by the unimodal auditory psychometric function). Note that the spatial reliability in the MCD model jointly depends on the stimulus and the receptive field of the input units; however, teasing apart the differential effects induced by these two sources of spatial uncertainty is beyond the scope of the present study. Hence, for simplicity, here we injected all spatial uncertainty into the stimulus (see also; *Parise and Ernst, 2016*; *Parise and Ernst, 2025*). For this simulation, we only used the data from observer LM of *Alais and Burr, 2004*, as it is the only participant for which the full dataset is publicly available (other observers, however, had similar results).

The stimuli are fed to the model to obtain the population response $MCD_{corr}(x,t)$ (see *Equation 6*), which is marginalized over time as follows.

$$\overline{MCD_{corr}}(x) = \sum_t MCD_{corr}(x,t) \tag{15}$$

This provides a distribution of the population response over the horizontal spatial dimension. Finally, a divisive normalization is performed to transform model response into a probability distribution.

$$p_{MCD}(x) = \frac{\overline{MCD_{MCD}}(x)}{\sum_x \overline{MCD_{MCD}}(x)} \tag{16}$$

It is important to note that *Equations 15; 16* have no free parameters, and all simulations are now performed with a fully constrained model. To test whether the MCD model can perform audiovisual integration according to the MLE model, we replicated the various conditions run by observer LM and calculated the bimodal likelihoods distribution predicted by the MLE and MCD models (i.e. $p_{MLE}(x)$ and $p_{MCD}(x)$). These were statistically identical, when considering rounding errors. The results of these simulations are plotted in *Figure 4C*, and displayed as cumulative distributions as in *Figure 1* of Alais and Burr, 2004.

Once demonstrated that $p_{MLE}(x) = p_{MCD}(x)$, it is clear that the MCD is equally capable of predicting observed responses. However, for completeness, we compared the prediction of the MCD to the empirical data: the results demonstrate that, just like the MLE model, a population of MCDs can predict both audiovisual bias (*Figure 4D*) and just noticeable differences (JND, *Figure 4E*). A MATLAB implementation of this simulation is included as *Source code 1*.

## MCD and BCI – simulation of *Körding et al., 2007*

To account for the spatial breakdown of multisensory integration, the BCI model operates in a hierarchical fashion: first, it estimates the probability that audiovisual stimuli share a common cause ($p(C = 1)$). Next, the model weighs and integrates the unimodal and bimodal information ($p_{mod}(x)$ and $p_{mle}(x)$) as follows:

$$p_{BCI,mod}(x) = p_{BCI}(C = 1) \cdot p_{MLE}(x) + [1 - p_{BCI}(C = 1)] \cdot p_{mod}(x) \tag{17}$$

All the terms in *Equation 17* have a clear analogue in the computations of the MCD population response: $p_{mle}(x)$ corresponds to $p_{MCD}(x)$ (see previous section). Likewise, the homologous of $p_{mod}(x)$ can be obtained by marginalizing over time and normalizing the output of the unimodal units as follows (see *Equation 15*)

$$\overline{MCD_{mod}}(x) = \frac{\sum_t MCD_{mod}(x, t)}{\sum_x \sum_t MCD_{mod}(x, t)} \tag{18}$$

Finally, the MCD homologous of BCI's $p(C = 1)$ can be read out from the population response following the same logic as *Equation 8*:

$$p_{MCD}(C = 1) = \Phi\left(\beta_{crit} + \beta_{corr} \log_{10} \sum_x \sum_t MCD_{corr}(x, t)\right) \tag{19}$$

Indeed, the output $\overline{MCD_{corr}}$ (*Equation 8*) not only decreases with increasing temporal disparity (see *Figure 1C*, left), but it also decreases with increasing spatial disparity (*Figure 4F*), thereby providing a measure of spatiotemporal coincidence that can be readily transformed into a probability for common cause (*Equation 19*). Here, we found that including a compressive non-linearity (logarithm of the total $MCD_{corr}$ response) provided a tighter fit to the empirical data. With $p_{MCD}(C = 1)$ representing the probability that vision and audition share a common cause, and $p_{MCD}(x)$ and $\overline{MCD_{mod}}(x)$ representing the bimodal and unimodal population responses, the MCD model can simulate observed responses as follows:

$$p_{MCD,mod}(x) = p_{MCD}(C = 1) \cdot p_{MCD}(x) + [1 - p_{MCD}(C = 1)] \cdot \overline{MCD_{mod}}(x) \tag{20}$$

The similarity between *Equation 20* and *Equation 17* demonstrates the fundamental homology of the BCI and MCD models: what remains to be tested is whether the MCD can also account for the results of *Körding et al., 2007*. For that, just like the BCI model, the MCD model also relies

on four free parameters: two are shared by both models, and represent the spatial uncertainty (i.e. the variance) of the unimodal input (i.e. $\sigma^2_{vid}$ and $\sigma^2_{aud}$). Additionally, the MCD model then needs two linear coefficients (slope $\beta_{corr}$ and intercept $\beta_{crit}$) to transform the dynamic population response into a probability of a common cause. Conversely, the remaining two parameters of the BCI model correspond to a prior for common cause and another for central location, neither of which are necessary to account for observed responses in the present framework. Parameters were fitted using BADS (*Acerbi and Ma, 2017*) set to maximize the Pearson correlation between model and human responses (*Figure 4G*). Overall, the MCD provided an excellent fit to the empirical data ($r=0.99$, *Figure 4G*), even slightly exceeding the performance of the BCI model while relying on the same degrees of freedom. Given that the fitted value of the slope parameter ($\beta_{corr}$) approached 1, we repeated the fitting while removing $\beta_{corr}$ from *Equation 19*: even with just three free parameters (one fewer than the BCI model), the MCD is in line with the BCI model and the correlation with the empirical data was 0.98. A Matlab implementation of this simulation is included as *Source code 1*.

## MCD and BCI – simulation of *Mohl et al., 2020*

Mohl and colleagues used eye movements to test whether humans and monkeys integrate audiovisual spatial cues according to BCI. The targets consisted of either unimodal or bimodal stimuli. Following the same logic as the simulation of *Alais and Burr, 2004* (see above), the unimodal input consisted of impulses with a Gaussian spatial profile (*Figure 4A–B*), whose the variance (i.e., $\sigma^2_{vid}$ and $\sigma^2_{aud}$) was set equal to the variance of the fixations measured in the unimodal trials (averaged across observers). Although the probability of a single fixation decreases with increasing disparity (*Figure 4H*, right), observers sometimes failed to make a second fixation even when the stimuli were noticeably far apart (lapses). This was especially true for monkeys, which had a lapse rate of 16%, indicative of low attention and compliance. To account for this, we can modify *Equation 19* and obtain the probability of a single fixation as follows:

$$p_{MCD}\left(C = 1\right) = p_{lapse} + \left(1 - 2p_{lapse}\right) \cdot \Phi\left(\beta_{crit} + \beta_{corr} \log_{10} \sum_x \sum_t MCD_{corr}\left(x, t\right)\right) \qquad (21)$$

Here, $p_{lapse}$ is a free parameter that represents the probability of making the incorrect number of fixations, irrespective of the discrepancy. As in *Equation 19*, $\beta_{corr}$ and $\beta_{crit}$ are free parameters that transform the dynamic population response into a probability of a common cause (i.e. single fixation). *Equation 21* could tightly reproduce the observed probability of a single fixation (*Figure 4H*, right), and the Pearson correlation between the model and data was 0.995 for monkeys and 0.988 for humans.

With the parameters $p_{lapse}$, $\beta_{corr}$ and $\beta_{crit}$ fitted to the probability of a single fixation (i.e. the probability of a common cause), it is now possible to predict gaze direction (i.e. the perceived location of the stimuli) with zero free parameters. For that, we can use *Equation 21* to get the probability of a common cause and predict gaze direction using *Equation 20*. The distribution of fixations predicted by the MCD closely follows the empirical histogram in both species (*Figure 4H*) and the correlation between the model and data was 0.9 for monkeys and 0.93 for humans. Note that *Figure 4H* shows only a subset of the 20 conditions tested in the experiment (the same subset of conditions shown in the original paper). A MATLAB implementation of this simulation is included as *Source code 1*.

## MCD and audiovisual gaze behavior

To test whether the dynamic MCD population response can account for gaze behavior during passive observation of audiovisual footage, a simple solution is to measure whether observers preferentially looked where the $MCD_{corr}$ response is maximal. Such an analysis was separately performed for each frame. For that, gaze directions were first low pass filtered with a Gaussian kernel ($\sigma = 14\,pixels$) and normalized to probabilities $p_{gaze}\left(x, y\right)$. Next, we calculated the average MCD responses at gaze direction for each frame (t); this was done by weighing $MCD_{corr}$ by $p_{gaze}$ as follows

$$\overline{MCD}_{gaze}(t) = \sum_x \sum_y p_{gaze}(x, y, t) \cdot MCD_{corr}(x, y, t) \qquad (22)$$

To assess the MCD response at gaze ($\overline{MCD_{gaze}}\,(t)$) is larger than the frame average, we calculated the standardized mean difference (SMD) for each frame as follows

$$SMD\,(t) = \frac{\overline{MCD_{gaze}}\,(t) - \mu_{x,y}\left[MCD_{corr}\,(x,y,t)\right]}{\sigma_{x,y}\left[MCD_{corr}\,(x,y,t)\right]} \tag{23}$$

Across the over 16,000 frames of the available dataset, the average SMD was 2.03 (*Figure 6D*). Given that the standardized mean difference serves as a metric for effect size, and that effect sizes surpassing 1.2 are deemed very large (*Sawilowsky, 2009*), it is remarkable that the MCD population model can so tightly account for human gaze behavior in a purely bottom-up fashion and without free parameters.

## Trimodal integration

Much like Bayesian ideal observer models, the present framework can be naturally extended to trimodal integration. As described in *Equation 6*, the, $MCD_{corr}\,(x,y,t)$ response is based on the pointwise product of input transients across modalities. In the bimodal case, this corresponds to the product of auditory and visual transient channels. For three modalities (e.g. auditory, visual, tactile), this generalizes to a trimodal coincidence detector, in which MCD units compute:

$$MCD_{trimodal}(x,y,t) = MCD_{aud}(x,y,t) \cdot \left[MCD_{aud}(x,y,t) * f_{lp}(t)\right] \cdot MCD_{vid}(x,y,t) \cdot \left[MCD_{vid}(x,y,t) * f_{lp}(t)\right] \cdot$$
$$MCD_{tac}(x,y,t) \cdot \left[MCD_{tac}(x,y,t) * f_{lp}(t)\right]$$

This detector responds maximally when transients in all three modalities co-occur in time and space. As in the bimodal case (*Figure 4*), the trimodal MCD response closely approximates the predictions of the maximum likelihood estimation (MLE) model (*Figure 4—figure supplement 1*).

$MCD_{lag}\,(x,y,t)$ (*Equation 7*) is instead defined via opponency (subtraction) between the two subunits of the MCD, which introduces a directional asymmetry between modalities. This structure makes it fundamentally pairwise. As a result, extending $MCD_{lag}\,(x,y,t)$ to three modalities would require computing pairwise lag estimates (AV, VT, AT) independently.

## Code availability statement

A variety MATLAB script running the MCD population model is included as *Source code 1*. This includes a code running the MCD on real-life footage (i.e., this is what was used to model MCD responses to the temporal determinants of multisensory integration, spatial orienting, and *Figure 5*). Additional codes simulate the experiment of *Alais and Carlile, 2005* and all simulations in *Figure 5*.

## Pre-processing of ecological audiovisual footage

The diversity of the stimuli used in this dataset requires some preprocessing before the stimuli can be fed to the population model. First, all movies were converted to grayscale (scaled between 0 and 1) and the soundtrack was converted to rms envelope (scaled between 0 and 1), thereby removing chromatic and tonal information. Movies were then padded with 2 s of frozen frames at onset and offset to accommodate for the manipulation of the lag. Finally, the luminance of the first still frame was set as baseline and subtracted from all subsequent frames (see Matlab codes in the *Source code 1*). Along with the padding, this helps minimizing transient artefacts induced by the onset of the video.

Video frames were scaled to 15% of the original size, and the static background was cropped. On a practical side, this made the simulations much faster (which is crucial for parameter estimation); on a theoretical side, such a down sampling simulates the Gaussian spatial pooling of luminance across the visual field (unfortunately, the present datasets do not provide sufficient information to convert pixels into visual angles). In a similar fashion, we down sampled sound envelope to match the frame rate of the video.

## Simulation of *Alais and Carlile, 2005*

For the simulations of *Horsfall et al., 2021*, the envelope of the auditory stimuli was extracted from the waveforms shown in the figures of the original publication. That was done using WebPlotDigitizer

to trace the profile of the waveforms; the digitized points were then interpolated and resampled at 1000 Hz. To preserve the manipulation of the direct-to-reverberant waves, the section of the envelope with the reverberant signal was identical across the four conditions (i.e. distances), so that what varied across conditions was the initial portion of the signals (the direct waves). For the simulations of *Horsfall et al., 2021*, all four psychometric functions were fitted simultaneously, so that the four psychometric functions all relied on just three free parameters: the ones related to the decision-making process. A MATLAB code running this simulation is now included as *Source code 1*.

## Individual observers' analysis

Most of the simulations described so far rely on group-level data, where psychometric curves represent the average response across the pool of observers that took part in each experiment. Individual psychometric functions, however, sometimes vary dramatically across observers; hence, one might wonder whether the MCD, besides predicting stimulus-driven variations in the psychometric functions, can also capture individual differences. A recent study by Yarrow and colleagues (*Yarrow et al., 2023*) directly addressed this question, and concluded that models of the Independent Channels family outperform the MCD at fitting responses individual differences.

Although it can be easily shown that such a conclusion was supported by an incomplete implementation of the MCD (which did not include the $MCD_{lag}$ output), a closer look at the two models against the same datasets help us illustrate their fundamental difference and highlight a key drawback of perceptual models that take parameters as input. Therefore, we first simulated the impulse stimuli used by Yarrow and colleagues (*Yarrow et al., 2023*), fed them to the MCD, and used *Equation 10* to generate the individual psychometric curves. Given that their stimuli consisted of temporal impulses with no spatiotemporal manipulation, a single MCD unit is sufficient to run these simulations. Overall, the model provided an excellent fit to the original data and tightly captured individual differences: the average Pearson correlation between predicted and empirical psychometric functions across the 57 curves shown in *Figure 2—figure supplement 3* is 0.98. Importantly, for such simulations, the MCD was fully constrained, and the only free parameters (3 in total) were the linear coefficients of *Equation 10*, which describe how the output of the MCD is used for perceptual decision-making. For comparison, also Independent Channels models achieved analogous goodness of fit, but they required at least five free parameters (depending on the exact implementation *Yarrow et al., 2023*).

To assess the generalizability of this finding, we additionally simulated the individual psychometric functions from the experiments that informed the architecture of the MCD units used here. Specifically, *Parise and Ernst, 2025* run two psychophysical studies using minimalistic stimuli that only varied over time. In the first one, auditory and visual stimuli consisted of step increments and/or decrements in intensity. Audiovisual lag was parametrically manipulated using the method of constant stimuli, and observers were required to perform both simultaneity and temporal order judgments. Following the logic described above, we fed the stimuli to the model, and used *Equation 10* (with three free parameters) to simulate human responses. Results demonstrate that the MCD can account for individual differences regardless of the task (*Figure 2—figure supplement 4*): the average Pearson correlation between empirical and predicted psychometric curves was 0.97 for the simultaneity judgments, and 0.96 for the temporal order judgments (64 individual psychometric curves from eight observers, for a total of 9600 trials). This generalizes the results of the previous simulation to a different type of stimuli (steps vs. impulses) and extends them to include a task, the temporal order judgment, which was not considered by Yarrow and colleagues (*Yarrow et al., 2023*) (whose model can only perform simultaneity judgments).

The second study of *Parise and Ernst, 2025* consists of simultaneity judgments for periodic audiovisual stimuli defined by a square-wave intensity envelope (*Figure 2—figure supplement 5*). Simultaneity judgments for this type of periodic stimuli are also periodic, with two complete oscillations in perceived simultaneity for each cycle of phase shifts between the senses (a phenomenon known as frequency doubling). Once again, using *Equation 10*, the MCD could account for individual differences in observed behavior (five psychometric curves from five observers, for a total of 3000 trials) with an average Pearson correlation of 0.93, while relying on just three free parameters.

It is important to note that for these simulations, the same MCD model accurately predicted (in a purely bottom-up fashion) bell-shaped SJ curves for non-periodic stimuli, and sinusoidal curves for periodic stimuli. Alternative models of audiovisual simultaneity that directly take lag as input always

enforce bell-shaped psychometric functions, where perceived synchrony monotonically decrease as we move away from the point of subjective simultaneity. As a result, in the absence of ad-hoc adjustments they all necessarily fail at replicating the results of Experiment 2 of *Parise and Ernst, 2025*, due to their inability to generate periodic psychometric functions. Conversely, the MCD is agnostic regarding the shape of the psychometric functions, hence the very same model used to predict the standard bell-shaped simultaneity judgments of *Yarrow et al., 2023* can also predict the periodic psychometric functions of *Parise and Ernst, 2025*, including individual differences across observers (all while relying on just three free parameters).

## Datasets

To thoroughly compare observed and model behavior, this study requires a large and diverse dataset consisting of both the raw stimuli and observers' responses. For that, we adopted a convenience sampling and simulated the studies for which both stimuli and responses were available (either in public repositories, shared by the authors, or extracted from published figures). The inclusion criteria depend on what aspect of multisensory integration is being investigated, and they are described below.

For the **temporal determinants of multisensory integration in humans**, we only included studies that: (1) used real-life audiovisual footage, (2) performed a parametric manipulation of lag, and (3) engaged observers in a forced-choice behavioral task. Forty-three individual experiments met the inclusion criteria (*Figure 2A-B*, *Figure 2—figure supplement 1*). These varied in terms of stimuli, observers, and tasks. In terms of stimuli, the dataset consists of responses from 105 unique real-life videos (see *Figure 2—figure supplement 1* and *Supplementary file 1*). The majority of the videos represented audiovisual speech (possibly the most common stimulus in audiovisual research), but they varied in terms of content (i.e. syllables, words, full sentences, etc.), intelligibility (i.e. sine-wave speech, amplitude-modulated noise, blurred visuals, etc.), composition (i.e. full face, mouth-only, oval frame, etc.), speaker identity, etc. The remaining non-speech stimuli consist of footage of actors playing a piano or a flute. The study from *Alais and Carlile, 2005* was included in the dataset because, even if the visual stimuli were minimalistic (blobs), the auditory stimuli consisted of ecological auditory depth cues depth cues recorded in a real reverberant environment (the Sydney Opera House, see below for details on the dataset). The dataset contains forced-choice responses to three different tasks: speech categorization (i.e., for the McGurk illusion), simultaneity judgments, and temporal order judgment. In terms of observers, besides the general population, the dataset consists of experimental groups varying in terms of age, musical expertise, and even includes a patient, PH, who reports hearing speech before seeing mouth movements after a lesion in the pons and basal ganglia (*Freeman et al., 2013*). Taken together, the dataset consists of ~1 k individual psychometric functions, from 454 unique observers, for a total of ~300 k trials; the psychometric curves for each experiment are shown in *Figure 2A-B*, *Figure 2—figure supplement 1*. All these simulations are based on psychometric functions averaged across observers, for simulations of individual observers, see *Figure 2—figure supplements 3–5*.

For the **temporal determinants of multisensory integration in rats**, we included studies that performed a parametric manipulation of lag and engaged rats in simultaneity and temporal order judgment tasks. Sixteen individual experiments (*Mafi et al., 2022*, *Schormans and Allman, 2018*, *Al Youzbaki et al., 2023*, *Schormans and Allman, 2023*, *Schormans et al., 2016*, *Paulcan et al., 2023*) met the inclusion criteria (*Figure 2C-D*, *Figure 2—figure supplement 2* and): all of them used minimalistic audiovisual stimuli (clicks and flashes) and with a parametric manipulation of audiovisual lag. Overall, the dataset consists of ~190 individual psychometric functions, from 110 rats (and 10 humans), for a total of ~300 k trials.

For the case of the **spatial determinants of multisensory integration**, to the best of our knowledge, there are no available datasets with both stimuli and psychophysical responses. Fortunately, however, the spatial aspects of multisensory integration are often studied with minimalistic audiovisual stimuli (e.g. clicks and blobs), which can be simulated exactly. Audiovisual integration in space is commonly framed in terms of optimal statistical estimation, where the bimodal percept is modelled either through MLE or BCI. To provide a plausible account for audiovisual integration in space, a

population of MCDs should also behave as a Bayesian-optimal estimator. This hypothesis was tested by comparing the population response to human data in the studies that originally tested the MLE and BCI models; hence, we simulated the study of *Alais and Burr, 2004* and *Körding et al., 2007*. Such simulations allow us to compare the data with our model, and our model with previous ones (MLE and BCI). Given that in these two simulations, a population of MCDs behaves just like the MLE and BCI models (and with the same number of free parameters or fewer), the current approach can be easily extended to other instances of sensory cue integration previously modelled in terms of optimal statistical estimation. This was tested by simulating the study of *Mohl et al., 2020*, who used eye movements to assess whether BCI can account for audiovisual integration in monkeys and humans (*Figure 4*).

Finally, we tested whether a population of MCDs can predict **audiovisual orienting and gaze behaviour** during passive observation of ecological audiovisual stimuli. *Coutrot and Guyader, 2015* run the ideal testbed for this hypothesis: much like our previous simulations (*Figure 2A*), they employed audiovisual speech stimuli, recorded indoor, with no camera shake. Specifically, they tracked eye movements from 20 observers who passively watched 15 videos of a lab meeting (see *Figure 6A*). Without fitting parameters, the population response tightly matched the empirical saliency maps (see *Figure 6B–D* and *Video 4*).

## MCD temporal filters: parameter estimation and generalizability

The temporal filters determine the temporal tuning of the model and consist of three parameters: two temporal constants of the unimodal band-pass filters ($\tau_{bpA}, \tau_{bpV}$; *Equation 1*), and one bimodal constant ($\tau_{lp}$, *Equation 5*). For humans, these parameter values were estimated by combining data from all 40 experiments shown in *Figure 2—figure supplement 1*, excluding the data from Patient PH. These experiments encompass SJ, TOJ, and McGurk tasks, each involving parametric manipulations of audiovisual lag, as well as tasks using ecological audiovisual stimuli (i.e. real-life footage). For rats, we followed the same approach, combining all psychometric curves shown in *Figure 2—figure supplement 2* (excluding pharmacological manipulation experiments), based on stimuli consisting of clicks and flashes in TOJ and SJ tasks.

To simulate each trial, we fed the stimuli into the population model to estimate internal response variables $\overline{MCD_{corr}}$ and $\overline{MCD_{lag}}$ (*Equations 8; 9*). Response probabilities were then derived from these internal signals as described in *Equation 10* (see also *Parise and Ernst, 2025*). This decision stage introduces three additional parameters: $\beta_{crit}$ (the criterion, a bias term), and $\beta_{corr}$, and $\beta_{lag}$ (gain parameters, acting as scaling factors). These decision-related parameters were estimated independently for each experiment. An exception was made for simulations based on *Schormans and Allman, 2018*; *Figure 2—figure supplement 2E*, which manipulated audio intensity across three levels—here, we used a single set of $\beta_{crit}$, $\beta_{corr}$, and $\beta_{lag}$ values across all conditions.

Two search algorithms were used for parameter estimation. First, we applied a global optimization method—Bayesian Adaptive Direct Search (BADS; Acerbi). To minimize the influence of initial values, we ran the BADS 200 times with different starting points (from 0.001 to 1.5 s). The best-fitting parameter values were then further refined using a local optimizer (fminsearch in MATLAB). The cost function minimized by both algorithms was 1 minus the Pearson correlation between real and simulated data, averaged across all experiments (each experiment was weighted equally, regardless of trial count or sample size). To assess robustness, we repeated the procedure using a mean squared error (MSE) cost function, which produced a set of parameter values ($\tau_{bpA}$=0.042 s $\tau_{bpV}$=0.039 s; $\tau_{lp}$=0.156 s) that are closely aligned with the original estimate ($\tau_{bpA}$=0.045 s $\tau_{bpV}$=0.036 s; $\tau_{lp}$=0.180 s).

This approach to parameter estimation, which aggregates data from a broad range of paradigms (n=40 for humans, n=15 for rats), enables us to estimate the temporal constants of the MCD units using a variety of ecological audiovisual stimuli (n=105 unique stimuli). Given that the stimuli consist of raw audiovisual footage, parameter estimation is computationally demanding — each search iteration can take up to 10 min. This makes standard techniques such as leave-one-out cross-validation impractical. The generalizability of the estimated temporal constants can nevertheless be tested against new, unseen data.

First, we tested whether the MCD model, with fixed temporal constants, could predict a well-known finding: that perceived audiovisual synchrony varies with distance in reverberant environments

(*Alais and Carlile, 2005*). To do this, we used the previously estimated temporal constants and fitted only a single set of three decision parameters to predict the four psychometric curves in *Figure 2B*—effectively using fewer than one free parameter per curve. Importantly, no changes were made to the model's temporal tuning. The MCD was able to fully reproduce the separation of the four curves across conditions. This result is not simply an extension of the model to a new stimulus set. Rather, it provides a purely bottom-up account of how perceived synchrony scales with spatial depth—without invoking any explicit computation of distance or higher-level inference mechanisms.

Next, we examined whether the estimated temporal constants—which had so far only been tested on group-averaged data—could also predict individual-level responses. To do this, we simulated individual observer data using previously published datasets: SJ data from *Yarrow et al., 2023*, and both TOJ and SJ data from *Parise and Ernst, 2025*, Experiments 1 and 2. These simulations allowed us to assess whether the model could generalize from group-level patterns to individual-level behaviour. The results confirmed that the MCD, using a single set of fixed temporal constants across all individuals, could successfully predict both individual and group data. Only the decision parameters ($\beta_{crit}$, $\beta_{corr}$, and $\beta_{lag}$) were fitted per observer, showing that temporal tuning itself can be considered constant across different observers.

Finally, using the same set of temporal constants, we evaluated whether the $\overline{MCD_{corr}}$ population output (*Equation 6*) could predict gaze behaviour during passive observation of real-life audiovisual footage—this time with zero free parameters. For this, we presented the audiovisual frames to the model (*Figure 6A*) and computed the MCD response for each pixel and time frame (*Figure 6B*). Across frames, the model's output consistently predicted the location of gaze direction: the region of the screen with the highest $\overline{MCD_{corr}}$ response systematically attracted participants' gaze (*Figure 6D*).

In summary, we have used data from a large pool of experiments consisting of 105 unique stimuli to estimate the temporal constants of the MCD model. These constants, once estimated, demonstrated predictive validity across a wide range of behavioural measures (SJ, TOJ, eye-tracking), stimulus types, species (humans and rats), and data granularities (group and individual). They generalized successfully to completely novel behavioural datasets and natural viewing conditions—including eye-tracking during passive video observation—all without any re-tuning of the model's core temporal filters.

## Acknowledgements

We thank Dr. Irene Senna for insightful comments and continuous support throughout all stages of this study. We are also grateful to Dr. Alessandro Moscatelli for valuable feedback on the manuscript, and to Prof. Marc Ernst for stimulating discussions during the early phases of this work.

## Additional information

### Funding
No external funding was received for this work.

### Author contributions
Cesare V Parise, Conceptualization, Resources, Data curation, Software, Formal analysis, Funding acquisition, Investigation, Visualization, Methodology, Writing – original draft, Project administration, Writing – review and editing

### Author ORCIDs
Cesare V Parise ⓘ https://orcid.org/0009-0000-6092-561X

Reviewer #1 (Public review): https://doi.org/10.7554/eLife.106122.3.sa1
Reviewer #2 (Public review): https://doi.org/10.7554/eLife.106122.3.sa2
Author response https://doi.org/10.7554/eLife.106122.3.sa3

# Additional files

## Supplementary files

MDAR checklist

Source code 1. This compressed folder includes Matlab codes and stimuli to simulate the experiments of *Alais and Burr, 2004*; *Alais and Carlile, 2005*; *Körding et al., 2007*; *Mohl et al., 2020*. Moreover, it also includes the code and video to replicate *Figure 5* and Video 3.

Supplementary file 1. Summary of the experiments simulated in *Figure 2—figure supplement 2*. The first column contains the reference of the study, the second column the task (McGurk, Simultaneity Judgment, and Temporal Order Judgment). The third column describes the stimuli: n represents the number of individual instances of the stimuli, 'HI' and 'LI' in *Magnotti and Beauchamp, 2017* indicate speech stimuli with High and Low Intelligibility, respectively. 'Blur' indicates that the videos were blurred. 'Disamb' indicates that ambiguous speech stimuli (i.e., sine-wave speech) were disambiguated by informing the observers of the original speech sound. The fourth column indicates whether visual and acoustic stimuli were congruent. Here, incongruent stimuli refer to the mismatching speech stimuli used in the McGurk task. 'SWS' indicates sine-wave speech; 'noise' in *Ikeda and Morishita, 2020* indicates a stimulus similar to sine-wave speech but in which white noise was used instead of pure sinusoidal waves. The fifth column represents the country where the study was performed. The sixth column describes the observers included in the study: 'c.s.' indicates convenience sampling (usually undergraduate students) musicians in *Lee and Noppeney, 2011*; *Lee and Noppeney, 2014* were amateur piano players; *Freeman et al., 2013* tested young observers (18–28 years old), a patient P.H. (67 years old) that after a lesion in the pons and basal ganglia reported hearing speech before seeing the lips move; and a group of age-matched controls (59–74 years old). The seventh column reports the number of observers included in the study. Overall, the full dataset consisted of 986 individual psychometric curves; however, several observers participated in more than one experiment, so that the total number of unique observers was 454. The eight column reports the number of lags used in the method of constant stimuli. The nineth column reports the number of trials included in the study. The tenth column reports the correlation between empirical and predicted psychometric functions. The bottom row contains some descriptive statistics of the dataset.

Supplementary file 2. Summary of the experiments simulated in *Figure 2—figure supplement 2*. The first column contains the reference of the study, the second column the task (Simultaneity Judgment and Temporal Order Judgment). The third column describes the stimuli. The fourth column indicates what rats were used as observers. The fifth column reports the number of rats in the study; 'same' means that the same rats took part in the experiment in the row above. The sixth column reports the number of lags used in the method of constant stimuli. The seventh column reports the number of trials included in the study (not available for all studies). The eighth column reports the correlation between empirical and predicted psychometric functions. The bottom row contains some descriptive statistics of the dataset.

## Data availability

No new data has been collected as part of this study. All datasets used here are publicly available and can be found here:- The tables in *Supplementary file 1* and *Supplementary file 2* contain all the details of the studies on the temporal constraints of audiovisual integration. That table includes the references to the original paper, where the data was either attached as a supplemetary file, or was directly grabbed from the figures in the original papers. For the spatial constraints of audiovisual integration, the data from *Alais and Burr, 2004* was grabbed from the figures in the original paper. For the experiment of *Körding et al., 2007*, the data has been shared by the authors and it is included with the Matlab files to show how the MCD model can be used to similate Bayesian Causal Inference. *Mohl et al., 2020* dataset available online: https://doi.org/10.5281/zenodo.3632106. *Coutrot and Guyader, 2015* dataset is available online: https://osf.io/kaqv2/overview.

The following previously published datasets were used:

| Author(s) | Year | Dataset title | Dataset URL | Database and Identifier |
|---|---|---|---|---|
| Mohl J | 2020 | jmohl/CI_behavioral: addressing reviews | https://doi.org/10.5281/zenodo.3900181 | Zenodo, 10.5281/zenodo.3900181 |
| Coutrot A, Guyader N | 2025 | Gaze - Conversation Scenes | https://osf.io/kaqv2/overview | Open Science Framework, kaqv2 |

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
