## [Editor Report · eLife Assessment]

This **important** study evaluates a model for multisensory correlation detection, focusing on the detection of correlated transients in visual and auditory stimuli. Overall, the experimental design is sound and the evidence is **compelling**. The synergy between the experimental and theoretical aspects of the article is strong, and the work will be of interest to both neuroscientists and psychologists working in the domain of sensory processing and perception

---

## [Referee Report · Reviewer #1 (Public review)]

Summary:

Parise presents another instantiation of the Multisensory Correlation Detector model that can now accept stimulus-level inputs. This is a valuable development as it removes researcher involvement in the characterization/labeling of features and allows analysis of complex stimuli with a high degree of nuance that was previously unconsidered (i.e. spatial/spectral distributions across time). The author demonstrates the power of the model by fitting data from dozens of previous experiments including multiple species, tasks, behavioral modality, and pharmacological interventions.

Strengths:

One of the model's biggest strengths, in my opinion, is its ability to extract complex spatiotemporal co-relationships from multisensory stimuli. These relationships have typically been manually computed or assigned based on stimulus condition and often distilled to a single dimension or even single number (e.g., "-50 ms asynchrony"). Thus, many models of multisensory integration depend heavily on human preprocessing of stimuli and these models miss out on complex dynamics of stimuli; the lead modality distribution apparent in figure 3b and c are provocative. I can imagine the model revealing interesting characteristics of the facial distribution of correlation during continuous audiovisual speech that have up to this point been largely described as "present" and almost solely focused on the lip area.

Another aspect that makes the MCD stand out among other models is the biological inspiration and generalizability across domains. The model was developed to describe a separate process - motion perception - and in a much simpler organism - drosophila. It could then describe a very basic neural computation that has been conserved across phylogeny (which is further demonstrated in the ability to predict rat, primate, and human data) and brain area. This aspect makes the model likely able to account for much more than what has already been demonstrated with only a few tweaks akin to the modifications described in this and previous articles from Parise.

What allows this potential is that, as Parise and colleagues have demonstrated in those papers since our (re)introduction of the model in 2016, the MCD model is modular - both in its ability to interface with different inputs/outputs and its ability to chain MCD units in a way that can analyze spatial, spectral, or any other arbitrary dimension of a stimulus. This fact leaves wide-open the possibilities for types of data, stimuli, and tasks a simplistic neutrally inspired model can account for.

And so it's unsurprising (but impressive!) that Parise has demonstrated the model's ability here to account for such a wide range of empirical data from numerous tasks (synchrony/temporal order judgement, localization, detection, etc.) and behavior types (manual/saccade responses, gaze, etc.) using only the stimulus and a few free parameters. This ability is another of the model's main strengths that I think deserves some emphasis: it represents a kind of validation of those experiments - especially in the context of cross-experiment predictions.

Finally, what is perhaps most impressive to me is that the MCD (and the accompanying decision model) does all this with very few (sometimes zero) free parameters. This highlights the utility of the model and the plausibility of its underlying architecture, but also helps to prevent extreme overfitting if fit correctly.

Weaknesses:

The model boasts an incredible versatility across tasks and stimulus configurations and its overall scope of the model is to understand how and what relevant sensory information is extracted from a stimulus. We still need to exercise care when interpreting its parameters, especially considering the broader context of top-down control of perception and that some multisensory mappings may not be derivable purely from stimulus statistics (e.g., the complementary nature of some phonemes/visemes).

---

## [Referee Report · Reviewer #2 (Public review)]

Summary:

Building on previous models of multisensory integration (including their earlier correlation-detection framework used for non-spatial signals), the author introduces a population-level Multisensory Correlation Detector (MCD) that processes raw auditory and visual data. Crucially, it does not rely on abstracted parameters, as is common in normative Bayesian models," but rather works directly on the stimulus itself (i.e., individual pixels and audio samples). By systematically testing the model against a range of experiments spanning human, monkey, and rat data - the authors show that their MCD population approach robustly predicts perception and behavior across species with a relatively small (0-4) number of free parameters.

Strengths:

(1) Unlike prior Bayesian models that used simplified or parameterized inputs, the model here is explicitly computable from full natural stimuli. This resolves a key gap in understanding how the brain might extract "time offsets" or "disparities" from continuously changing audio-visual streams.

(2) The same population MCD architecture captures a remarkable range of multisensory phenomena, from classical illusions (McGurk, ventriloquism) and synchrony judgments, to attentional/gaze behavior driven by audio-visual salience. This generality strongly supports the idea that a single low-level computation (correlation detection) can underlie many distinct multisensory effects.

(3) By tuning model parameters to different temporal rhythms (e.g., faster in rodents, slower in humans), the MCD explains cross-species perceptual data without reconfiguring the underlying architecture.

(4) The authors frame their model as a plausible algorithmic account of the Bayesian multisensory-integration models in Marr's levels of hierarchy.

Weaknesses:

What remains unclear is how the parameters themselves relate to stimulus quantities (like stimulus uncertainty), as is often straightforward in Bayesian models. A theoretical missing link is the explicit relationship between the parameters of the MCD models and those of a cue combination model, thereby bridging Marr's levels of hierarchy.

Likely Impact and Usefulness

The work offers a compelling unification of multiple multisensory tasks-temporal order judgments, illusions, Bayesian causal inference, and overt visual attention-under a single, fully stimulus-driven framework. Its success with natural stimuli should interest computational neuroscientists, systems neuroscientists, and machine learning scientists. This paper thus makes an important contribution to the field by moving beyond minimalistic lab stimuli, illustrating how raw audio and video can be integrated using elementary correlation analyses.

---

## [Author Response]

The following is the authors’ response to the original reviews.

**Reviewer #1 (Public review):**
Summary:Parise presents another instantiation of the Multisensory Correlation Detector model that can now accept stimulus-level inputs. This is a valuable development as it removes researcher involvement in the characterization/labeling of features and allows analysis of complex stimuli with a high degree of nuance that was previously unconsidered (i.e., spatial/spectral distributions across time). The author demonstrates the power of the model by fitting data from dozens of previous experiments, including multiple species, tasks, behavioral modalities, and pharmacological interventions.

Thanks for the kind words!

Strengths:One of the model's biggest strengths, in my opinion, is its ability to extract complex spatiotemporal co-relationships from multisensory stimuli. These relationships have typically been manually computed or assigned based on stimulus condition and often distilled to a single dimension or even a single number (e.g., "-50 ms asynchrony"). Thus, many models of multisensory integration depend heavily on human preprocessing of stimuli, and these models miss out on complex dynamics of stimuli; the lead modality distribution apparent in Figures 3b and c is provocative. I can imagine the model revealing interesting characteristics of the facial distribution of correlation during continuous audiovisual speech that have up to this point been largely described as "present" and almost solely focused on the lip area.Another aspect that makes the MCD stand out among other models is the biological inspiration and generalizability across domains. The model was developed to describe a separate process - motion perception - and in a much simpler organism - Drosophila. It could then describe a very basic neural computation that has been conserved across phylogeny (which is further demonstrated in the ability to predict rat, primate, and human data) and brain area. This aspect makes the model likely able to account for much more than what has already been demonstrated with only a few tweaks akin to the modifications described in this and previous articles from Parise.What allows this potential is that, as Parise and colleagues have demonstrated in those papers since our (re)introduction of the model in 2016, the MCD model is modular - both in its ability to interface with different inputs/outputs and its ability to chain MCD units in a way that can analyze spatial, spectral, or any other arbitrary dimension of a stimulus. This fact leaves wide open the possibilities for types of data, stimuli, and tasks a simplistic, neutrally inspired model can account for.And so it's unsurprising (but impressive!) that Parise has demonstrated the model's ability here to account for such a wide range of empirical data from numerous tasks (synchrony/temporal order judgement, localization, detection, etc.) and behavior types (manual/saccade responses, gaze, etc.) using only the stimulus and a few free parameters. This ability is another of the model's main strengths that I think deserves some emphasis: it represents a kind of validation of those experiments, especially in the context of cross-experiment predictions (but see some criticism of that below).Finally, what is perhaps most impressive to me is that the MCD (and the accompanying decision model) does all this with very few (sometimes zero) free parameters. This highlights the utility of the model and the plausibility of its underlying architecture, but also helps to prevent extreme overfitting if fit correctly (but see a related concern below).

We sincerely thank the reviewer for their thoughtful and generous comments. We are especially pleased that the core strengths of the model—its stimulus-computable architecture, biological grounding, modularity, and cross-domain applicability—were clearly recognized. As the reviewer rightly notes, removing researcher-defined abstractions and working directly from naturalistic stimuli opens the door to uncovering previously overlooked dynamics in complex multisensory signals, such as the spatial and temporal richness of audiovisual speech.

We also appreciate the recognition of the model’s origins in a simple organism and its generalization across species and behaviors. This phylogenetic continuity reinforces our view that the MCD captures a fundamental computation with wide-ranging implications. Finally, we are grateful for the reviewer’s emphasis on the model’s predictive power across tasks and datasets with few or no free parameters—a property we see as key to both its parsimony and explanatory utility.

We have highlighted these points more explicitly in the revised manuscript, and we thank the reviewer for their generous and insightful endorsement of the work.

Weaknesses:There is an insufficient level of detail in the methods about model fitting. As a result, it's unclear what data the models were fitted and validated on. Were models fit individually or on average group data? Each condition separately? Is the model predictive of unseen data? Was the model cross-validated? Relatedly, the manuscript mentions a randomization test, but the shuffled data produces model responses that are still highly correlated to behavior despite shuffling. Could it be that any stimulus that varies in AV onset asynchrony can produce a psychometric curve that matches any other task with asynchrony judgements baked into the task? Does this mean all SJ or TOJ tasks produce correlated psychometric curves? Or more generally, is Pearson's correlation insensitive to subtle changes here, considering psychometric curves are typically sigmoidal? Curves can be non-overlapping and still highly correlated if one is, for example, scaled differently. Would an error term such as mean-squared or root mean-squared error be more sensitive to subtle changes in psychometric curves? Alternatively, perhaps if the models aren't cross-validated, the high correlation values are due to overfitting?

The reviewer is right: the current version of the manuscript only provides limited information about parameter fitting. In the revised version of the manuscript, we included a parameter estimation and generalizability section that includes all information requested by the reviewer.

To test whether using the MSE instead of Pearson correlation led to a similar estimated set of parameter values, we repeated the fitting using the MSE. The parameter estimated with this method (TauV, TauA, TauBim) closely followed those estimated using Pearson correlation (TauV, TauA, TauBim). Given the similarity of these results, we have chosen not to include further figures, however this analysis is now included in the new section (pages 23-24).

Regarding the permutation test, it is expected that different stimuli produce analogous psychometric functions: after all, all studies relied on stimuli containing identical manipulation of lags. As a result, MCD population responses tend to be similar across experiments. Therefore, it is not a surprise that the permuted distribution of MCD-data correlation in Supplementary Figure 1K has a mean as high as 0.97. However, what is important is to demonstrate that the non-permuted dataset has an even higher goodness of fit. Supplementary Figure 1K demonstrates that none of the permuted stimuli could outperform the non-permuted dataset; the mean of the non-permuted distribution is 4.7 (standard deviations) above the mean of the already high permuted distribution.

We believe the new section, along with the present response, fully addresses the legitimate concerns of the reviewer.

While the model boasts incredible versatility across tasks and stimulus configurations, fitting behavioral data well doesn't mean we've captured the underlying neural processes, and thus, we need to be careful when interpreting results. For example, the model produces temporal parameters fitting rat behavior that are 4x faster than when fitting human data. This difference in slope and a difference at the tails were interpreted as differences in perceptual sensitivity related to general processing speeds of the rat, presumably related to brain/body size differences. While rats no doubt have these differences in neural processing speed/integration windows, it seems reasonable that a lot of the differences in human and rat psychometric functions could be explained by the (over)training and motivation of rats to perform on every trial for a reward - increasing attention/sensitivity (slope) - and a tendency to make mistakes (compression evident at the tails). Was there an attempt to fit these data with a lapse parameter built into the decisional model as was done in Equation 21? Likewise, the fitted parameters for the pharmacological manipulations during the SJ task indicated differences in the decisional (but not the perceptual) process and the article makes the claim that "all pharmacologically-induced changes in audiovisual time perception" can be attributed to decisional processes "with no need to postulate changes in low-level temporal processing." However, those papers discuss actual sensory effects of pharmacological manipulation, with one specifically reporting changes to response timing. Moreover, and again contrary to the conclusions drawn from model fits to those data, both papers also report a change in psychometric slope/JND in the TOJ task after pharmacological manipulation, which would presumably be reflected in changes to the perceptual (but not the decisional) parameters.

Fitting or predicting behaviour does not in itself demonstrate that a model captures the underlying neural computations—though it may offer valuable constraints and insights. In line with this, we were careful not to extrapolate the implications of our simulations to specific neural mechanisms.

Temporal sensitivity is, by definition, a behavioural metric, and—as the reviewer correctly notes—its estimation may reflect a range of contributing factors beyond low-level sensory processing, including attention, motivation, and lapse rates (i.e., stimulus-independent errors). In Equation 21, we introduced a lapse parameter specifically to account for such effects in the context of monkey eye-tracking data. For the rat datasets, however, the inclusion of a lapse term was not required to achieve a close fit to the psychometric data (ρ = 0.981). While it is likely that adding a lapse component would yield a marginally better fit, the absence of single-trial data prevents us from applying model comparison criteria such as AIC or BIC to justify the additional parameter. In light of this, and to avoid unnecessary model complexity, we opted not to include a lapse term in the rat simulations.

With respect to the pharmacological manipulation data, we acknowledge the reviewer’s point that observed changes in slope and bias could plausibly arise from alterations at either the sensory or decisional level—or both. In our model, low-level sensory processing is instantiated by the MCD architecture, which outputs the MCDcorr and MCDlag signals that are then scaled and integrated during decision-making. Importantly, this scaling operation influences the slope of the resulting psychometric functions, such that changes in slope can arise even in the absence of any change to the MCD’s temporal filters. In our simulations, the temporal constants of the MCD units were fixed to the values estimated from the non-pharmacological condition (see parameter estimation section above), and only the decision-related parameters were allowed to vary. From this modelling perspective, the behavioural effects observed in the pharmacological datasets can be explained entirely by changes at the decisional level. However, we do not claim that such an explanation excludes the possibility of genuine sensory-level changes. Rather, we assert that our model can account for the observed data without requiring modifications to early temporal tuning.

To rigorously distinguish sensory from decisional effects, future experiments will need to employ stimuli with richer temporal structure—e.g., temporally modulated sequences of clicks and flashes that vary in frequency, phase, rhythm, or regularity (see Fujisaki & Nishida, 2007; Denison et al., 2012; Parise & Ernst, 2016, 2025; Locke & Landy, 2017; Nidiffer et al., 2018). Such stimuli engage the MCD in a more stimulus-dependent manner, enabling a clearer separation between early sensory encoding and later decision-making processes. Unfortunately, the current rat datasets—based exclusively on single click-flash pairings—lack the complexity needed for such disambiguation. As a result, while our simulations suggest that the observed pharmacologically induced effects can be attributed to changes in decision-level parameters, they do not rule out concurrent sensory-level changes.

In summary, our results indicate that changes in the temporal tuning of MCD units are not necessary to reproduce the observed pharmacological effects on audiovisual timing behaviour. However, we do not assert that such changes are absent or unnecessary in principle. Disentangling sensory and decisional contributions will ultimately require richer datasets and experimental paradigms designed specifically for this purpose. We have now modified the results section (page 6) and the discussion (page 11) to clarify these points.

The case for the utility of a stimulus-computable model is convincing (as I mentioned above), but its framing as mission-critical for understanding multisensory perception is overstated, I think. The line for what is "stimulus computable" is arbitrary and doesn't seem to be followed in the paper. A strict definition might realistically require inputs to be, e.g., the patterns of light and sound waves available to our eyes and ears, while an even more strict definition might (unrealistically) require those stimuli to be physically present and transduced by the model. A reasonable looser definition might allow an "abstract and low-dimensional representation of the stimulus, such as the stimulus envelope (which was used in the paper), to be an input. Ultimately, some preprocessing of a stimulus does not necessarily confound interpretations about (multi)sensory perception. And on the flip side, the stimulus-computable aspect doesn't necessarily give the model supreme insight into perception. For example, the MCD model was "confused" by the stimuli used in our 2018 paper (Nidiffer et al., 2018; Parise & Ernst, 2025). In each of our stimuli (including catch trials), the onset and offset drove strong AV temporal correlations across all stimulus conditions (including catch trials), but were irrelevant to participants performing an amplitude modulation detection task. The to-be-detected amplitude modulations, set at individual thresholds, were not a salient aspect of the physical stimulus, and thus only marginally affected stimulus correlations. The model was of course, able to fit our data by "ignoring" the on/offsets (i.e., requiring human intervention), again highlighting that the model is tapping into a very basic and ubiquitous computational principle of (multi)sensory perception. But it does reveal a limitation of such a stimulus-computable model: that it is (so far) strictly bottom-up.

We appreciate the reviewer’s thoughtful engagement with the concept of stimulus computability. We agree that the term requires careful definition and should not be taken as a guarantee of perceptual insight or neural plausibility. In our work, we define a model as “stimulus-computable” if all its inputs are derived directly from the stimulus, rather than from experimenter-defined summary descriptors such as temporal lag, spatial disparity, or cue reliability. In the context of multisensory integration, this implies that a model must account not only for how cues are combined, but also for how those cues are extracted from raw inputs—such as audio waveforms and visual contrast sequences.

This distinction is central to our modelling philosophy. While ideal observer models often specify how information should be combined once identified, they typically do not address the upstream question of how this information is extracted from sensory input. In that sense, models that are not stimulus-computable leave out a key part of the perceptual pipeline. We do not present stimulus computability as a marker of theoretical superiority, but rather as a modelling constraint that is necessary if one’s aim is to explain how structured sensory input gives rise to perception. This is a view that is also explicitly acknowledged and supported by Reviewer 2.

Framed in Marr’s (1982) terms, non–stimulus-computable models tend to operate at the computational level, defining what the system is doing (e.g., computing a maximum likelihood estimate), whereas stimulus-computable models aim to function at the algorithmic level, specifying how the relevant representations and operations might be implemented. When appropriately constrained by biological plausibility, such models may also inform hypotheses at the implementational level, pointing to potential neural substrates that could instantiate the computation.

Regarding the reviewer’s example illustrating a limitation of the MCD model, we respectfully note that the account appears to be based on a misreading of our prior work. In Parise & Ernst (2025), where we simulated the stimuli from Nidiffer et al. (2018), the MCD model reproduced participants’ behavioural data without any human intervention or adjustment. The model was applied in a fully bottom-up, stimulus-driven manner, and its output aligned with observer responses as-is. We suspect the confusion may stem from analyses shown in Figure 6 - Supplement Figure 5 of Parise & Ernst (2025), where we investigated the lack of a frequency-doubling effect in the Nidiffer et al. data. However, those analyses were based solely on the Pearson correlation between auditory and visual stimulus envelopes and did not involve the MCD model. No manual exclusion of onset/offset events was applied, nor was the MCD used in those particular figures. We also note that Parise & Ernst (2025) is a separate, already published study and is not the manuscript currently under review.

In summary, while we fully agree that stimulus computability does not resolve all the complexities of multisensory perception (see comments below about speech), we maintain that it provides a valuable modelling constraint—one that enables robust, generalisable predictions when appropriately scoped.

The manuscript rightly chooses to focus a lot of the work on speech, fitting the MCD model to predict behavioral responses to speech. The range of findings from AV speech experiments that the MCD can account for is very convincing. Given the provided context that speech is "often claimed to be processed via dedicated mechanisms in the brain," a statement claiming a "first end-to-end account of multisensory perception," and findings that the MCD model can account for speech behaviors, it seems the reader is meant to infer that energetic correlation detection is a complete account of speech perception. I think this conclusion misses some facets of AV speech perception, such as integration of higher-order, non-redundant/correlated speech features (Campbell, 2008) and also the existence of top-down and predictive processing that aren't (yet!) explained by MCD. For example, one important benefit of AV speech is interactions on linguistic processes - how complementary sensitivity to articulatory features in the auditory and visual systems (Summerfield, 1987) allow constraint of linguistic processes (Peelle & Sommers, 2015; Tye-Murray et al., 2007).

We thank the reviewer for their thoughtful comments, and especially for the kind words describing the range of findings from our AV speech simulations as “very convincing.”

We would like to clarify that it is not our view that speech perception can be reduced to energetic correlation detection. While the MCD model captures low- to mid-level temporal dependencies between auditory and visual signals, we fully agree that a complete account of audiovisual speech perception must also include higher-order processes—including linguistic mechanisms and top-down predictions. These are critical components of AV speech comprehension, and lie beyond the scope of the current model.

Our use of the term “end-to-end” is intended in a narrow operational sense: the model transforms raw audiovisual input (i.e., audio waveforms and video frames) directly into behavioural output (i.e., button press responses), without reliance on abstracted stimulus parameters such as lag, disparity or reliability. It is in this specific technical sense that the MCD offers an end-to-end model. We have revised the manuscript to clarify this usage to avoid any misunderstanding.

In light of the reviewer’s valuable point, we have now edited the Discussion to acknowledge the importance of linguistic processes (page 13) and to clarify what we mean by end-to-end account (page 11). We agree that future work will need to explore how stimulus-computable models such as the MCD can be integrated with broader frameworks of linguistic and predictive processing (e.g., Summerfield, 1987; Campbell, 2008; Peelle & Sommers, 2015; Tye-Murray et al., 2007).

References

Campbell, R. (2008). The processing of audio-visual speech: empirical and neural bases. Philosophical Transactions of the Royal Society B: Biological Sciences, 363(1493), 1001-1010. https://doi.org/10.1098/rstb.2007.2155

Nidiffer, A. R., Diederich, A., Ramachandran, R., & Wallace, M. T. (2018). Multisensory perception reflects individual differences in processing temporal correlations. Scientific Reports 2018 8:1, 8(1), 1-15. https://doi.org/10.1038/s41598-018-32673-y

Parise, C. V, & Ernst, M. O. (2025). Multisensory integration operates on correlated input from unimodal transient channels. ELife, 12. https://doi.org/10.7554/ELIFE.90841

Peelle, J. E., & Sommers, M. S. (2015). Prediction and constraint in audiovisual speech perception. Cortex, 68, 169-181. https://doi.org/10.1016/j.cortex.2015.03.006

Summerfield, Q. (1987). Some preliminaries to a comprehensive account of audio-visual speech perception. In B. Dodd & R. Campbell (Eds.), Hearing by Eye: The Psychology of Lip-Reading (pp. 3-51). Lawrence Erlbaum Associates.

Tye-Murray, N., Sommers, M., & Spehar, B. (2007). Auditory and Visual Lexical Neighborhoods in Audiovisual Speech Perception: Trends in Amplification, 11(4), 233-241. https://doi.org/10.1177/1084713807307409

**Reviewer #2 (Public review):**
Summary:Building on previous models of multisensory integration (including their earlier correlation-detection framework used for non-spatial signals), the author introduces a population-level Multisensory Correlation Detector (MCD) that processes raw auditory and visual data. Crucially, it does not rely on abstracted parameters, as is common in normative Bayesian models," but rather works directly on the stimulus itself (i.e., individual pixels and audio samples). By systematically testing the model against a range of experiments spanning human, monkey, and rat data, the authors show that their MCD population approach robustly predicts perception and behavior across species with a relatively small (0-4) number of free parameters.Strengths:(1) Unlike prior Bayesian models that used simplified or parameterized inputs, the model here is explicitly computable from full natural stimuli. This resolves a key gap in understanding how the brain might extract "time offsets" or "disparities" from continuously changing audio-visual streams.(2) The same population MCD architecture captures a remarkable range of multisensory phenomena, from classical illusions (McGurk, ventriloquism) and synchrony judgments, to attentional/gaze behavior driven by audio-visual salience. This generality strongly supports the idea that a single low-level computation (correlation detection) can underlie many distinct multisensory effects.(3) By tuning model parameters to different temporal rhythms (e.g., faster in rodents, slower in humans), the MCD explains cross-species perceptual data without reconfiguring the underlying architecture.

We thank the reviewer for their positive evaluation of the manuscript, and particularly for highlighting the significance of the model's stimulus-computable architecture and its broad applicability across species and paradigms. Please find our responses to the individual points below.

Weaknesses:(1) The authors show how a correlation-based model can account for the various multisensory integration effects observed in previous studies. However, a comparison of how the two accounts differ would shed light on the correlation model being an implementation of the Bayesian computations (different levels in Marr's hierarchy) or making testable predictions that can distinguish between the two frameworks. For example, how uncertainty in the cue combined estimate is also the harmonic mean of the unimodal uncertainties is a prediction from the Bayesian model. So, how the MCD framework predicts this reduced uncertainty could be one potential difference (or similarity) to the Bayesian model.

We fully agree with the reviewer that a comparison between the correlation-based MCD model and Bayesian accounts is valuable—particularly for clarifying how the two frameworks differ conceptually and where they may converge.

As noted in the revised manuscript, the key distinction lies in the level of analysis described by Marr (1982). Bayesian models operate at the computational level, describing what the system is aiming to compute (e.g., optimal cue integration). In contrast, the MCD functions at the algorithmic level, offering a biologically plausible mechanism for how such integration might emerge from stimulus-driven representations.

In this context, the MCD provides a concrete, stimulus-grounded account of how perceptual estimates might be constructed—potentially implementing computations with Bayesian-like characteristics (e.g., reduced uncertainty, cue weighting). Thus, the two models are not mutually exclusive but can be seen as complementary: the MCD may offer an algorithmic instantiation of computations that, at the abstract level, resemble Bayesian inference.

We have now updated the manuscript to explicitly highlight this relationship (pages 2 and 11). In the revised manuscript, we also included a new figure (Figure 5) and movie (Supplementary Movie 3), to show how the present approach extends previous Bayesian models for the case of cue integration (i.e., the ventriloquist effect).

(2) The authors show a good match for cue combination involving 2 cues. While Bayesian accounts provide a direction for extension to more cues (also seen empirically, for eg, in Hecht et al. 2008), discussion on how the MCD model extends to more cues would benefit the readers.

We thank the reviewer for this insightful comment: extending the MCD model to include more than two sensory modalities is a natural and valuable next step. Indeed, one of the strengths of the MCD framework lies in its modularity. Let us consider the MCDcorr output (Equation 6), which is computed as the pointwise product of transient inputs across modalities. Extending this to include a third modality, such as touch, is straightforward: MCD units would simply multiply the transient channels from all three modalities, effectively acting as trimodal coincidence detectors that respond when all inputs are aligned in time and space.

By contrast, extending MCDlag is less intuitive, due to its reliance on opponency between two subunits (via subtraction). A plausible solution is to compute MCDlag in a pairwise fashion (e.g., AV, VT, AT), capturing relative timing across modality pairs.

Importantly, the bulk of the spatial integration in our framework is carried by MCDcorr, which generalises naturally to more than two modalities. We have now formalised this extension and included a graphical representation in a supplementary section of the revised manuscript.

Likely Impact and Usefulness:The work offers a compelling unification of multiple multisensory tasks- temporal order judgments, illusions, Bayesian causal inference, and overt visual attention - under a single, fully stimulus-driven framework. Its success with natural stimuli should interest computational neuroscientists, systems neuroscientists, and machine learning scientists. This paper thus makes an important contribution to the field by moving beyond minimalistic lab stimuli, illustrating how raw audio and video can be integrated using elementary correlation analyses.
**Reviewer #1 (Recommendations for the authors):**
Recommendations:My biggest concern is a lack of specificity about model fitting, which is assuaged by the inclusion of sufficient detail to replicate the analysis completely or the inclusion of the analysis code. The code availability indicates a script for the population model will be included, but it is unclear if this code will provide the fitting details for the whole of the analysis.

We thank the reviewer for raising this important point. A new methodological section has been added to the manuscript, detailing the model fitting procedures used throughout the study. In addition, the accompanying code repository now includes MATLAB scripts that allow full replication of the spatiotemporal MCD simulations.

Perhaps it could be enlightening to re-evaluate the model with a measure of error rather than correlation? And I think many researchers would be interested in the model's performance on unseen data.

The model has now been re-evaluated using mean squared error (MSE), and the results remain consistent with those obtained using Pearson correlation. Additionally, we have clarified which parts of the study involve testing the model on unseen data (i.e., data not used to fit the temporal constants of the units). These analyses are now included and discussed in the revised fitting section of the manuscript (pages 23-24).

Otherwise, my concerns involve the interpretation of findings, and thus could be satisfied with minor rewording or tempering conclusions.

The manuscript has been revised to address these interpretative concerns, with several conclusions reworded or tempered accordingly. All changes are marked in blue in the revised version.

Miscellanea:Should b0 in equation 10 be bcrit to match the below text?

Thank you for catching this inconsistency. We have corrected Equation 10 (and also Equation 21) to use the more transparent notation bcrit instead of b0, in line with the accompanying text.

Equation 23, should time be averaged separately? For example, if multiple people are speaking, the average correlation for those frames will be higher than the average correlation across all times.

We thank the reviewer for raising this thoughtful and important point. In response, we have clarified the notation of Equation 23 in the revised manuscript (page 20). Specifically, we now denote the averaging operations explicitly as spatial means and standard deviations across all pixel locations within each frame.

This equation computes the z-score of the MCD correlation value at the current gaze location, normalized relative to the spatial distribution of correlation values in the same frame. That is, all operations are performed at the frame level, not across time. This ensures that temporally distinct events are treated independently and that the final measure reflects relative salience within each moment, not a global average over the stimulus. In other words, the spatial distribution of MCD activity is re-centered and rescaled at each frame, exactly to avoid the type of inflation or confounding the reviewer rightly cautioned against.

**Reviewer #2 (Recommendations for the authors):**
The authors have done a great job of providing a stimulus computable model of cue combination. I had just a few suggestions to strengthen the theoretical part of the paper:(1) While the authors have shown a good match between MCD and cue combination, some theoretical justification or equivalence analysis would benefit readers on how the two relate to each other. Something like Zhang et al. 2019 (which is for motion cue combination) would add to the paper.

We agree that it is important to clarify the theoretical relationship between the Multisensory Correlation Detector (MCD) and normative models of cue integration, such as Bayesian combination. In the revised manuscript, we have now modified the introduction and added a paragraph in the Discussion addressing this link more explicitly. In brief, we see the MCD as an algorithmic-level implementation (in Marr’s terms) that may approximate or instantiate aspects of Bayesian inference.

(2) Simulating cue combination for tasks that require integration of more than two cues (visual, auditory, haptic cues) would more strongly relate the correlation model to Bayesian cue combination. If that is a lot of work, at least discussing this would benefit the paper

This point has now been addressed, and a new paragraph discussing the extension of the MCD model to tasks involving more than two sensory modalities has been added to the Discussion section.